# Possibilistic Predictive Uncertainty for Deep Learning

Yao Ni [1]   Jeremie Houssineau [1]   Yew-Soon Ong [1,2]   Piotr Koniusz [3,4]

## Abstract

Deep neural networks achieve impressive results across diverse applications, yet their overconfidence on unseen inputs necessitates reliable epistemic uncertainty modeling. Existing methods for uncertainty modeling face a fundamental dilemma: Bayesian approaches provide principled estimates but remain computationally prohibitive, while efficient second-order predictors lack rigorous connections between their specific objectives and epistemic uncertainty quantification. To resolve this dilemma, we introduce **D**irichlet-**a**pproximated **p**ossibilistic **p**osterior p**r**edictions (DAPPr), a principled framework grounded in possibility theory. We define a possibilistic posterior over parameters, project it to the prediction space via supremum operators, and approximate the projected posterior using learnable Dirichlet possibility functions. This projection-and-approximation strategy yields a simple training objective with closed-form solutions. Despite its simplicity, extensive experiments across diverse benchmarks show that DAPPr achieves competitive or superior uncertainty quantification performance over state-of-the-art second-order predictors while maintaining both principled derivation and computational efficiency. Code is available at https://github.com/MaxwellYaoNi/DAPPr.

## 1. Introduction

Recent breakthroughs in deep neural nets (He et al., 2016; Dosovitskiy et al., 2021) are primarily driven by training expressive models (Kaplan et al., 2020) on massive datasets (Deng et al., 2009; Kuznetsova et al., 2020; Schuhmann et al., 2022). This data-driven paradigm has achieved remarkable success across diverse computer vision tasks from image classification (He et al., 2016; Kirillov et al., 2023;

Wang et al., 2024; Zhang et al., 2024) to image generation (Rombach et al., 2022; Ni et al., 2022; Peebles & Xie, 2023).

Despite their impressive success, deep models are prone to overconfidence (Nguyen et al., 2015; Mehrtash et al., 2020; Zhang et al., 2025), assigning high confidence to incorrect predictions, particularly on data beyond their training distribution. Such overconfidence becomes critical in high-stakes applications such as autonomous driving (Wang et al., 2021) and medical diagnosis (Mehrtash et al., 2020), and in security-sensitive scenarios (Dong et al., 2026; Yu et al., 2026), where incorrect predictions can be catastrophic. To ensure safe deployment, models must recognize unreliable predictions. This requires quantifying *predictive uncertainty*, the overall uncertainty associated with a prediction, which originates from two sources: *aleatoric uncertainty*, intrinsic to data ambiguity and irreducible, and *epistemic uncertainty*, due to limited knowledge and reducible with more data (Kendall & Gal, 2017). Since aleatoric uncertainty is fixed by the data, the only remaining path to improving reliability lies in reducing epistemic uncertainty, yet standard deep models lack an explicit mechanism to represent it.

A principled approach to representing epistemic uncertainty is Bayesian learning, which characterizes uncertainty through the disagreement among multiple models that explain the observed data well. Formalizing this disagreement requires marginalizing over the posterior distribution of parameters, which is computationally intractable and necessitates approximations such as Bayesian neural networks (Blundell et al., 2015; Krueger et al., 2017; Jospin et al., 2022; Rudner et al., 2022), MC Dropout (Gal & Ghahramani, 2016; Ovadia et al., 2019; Mobiny et al., 2019), and ensemble methods (Lakshminarayanan et al., 2017; Wen et al., 2020; Löhr et al., 2025). While theoretically appealing, these approximations are computationally expensive and challenging to scale to large networks and datasets (Liu et al., 2020; He et al., 2020; Mukhoti et al., 2023).

To avoid the computational burden of Bayesian marginalization, an alternative line of work models distributions over output predictions rather than over parameters. By representing predictions as distributions rather than point estimates, these methods form second-order representations that capture epistemic uncertainty through distributional properties such as concentration or dispersion. Such representations

[1]Nanyang Technological University [2]A*STAR [3]The University of New South Wales [4]Data61♥CSIRO. Correspondence to: Jeremie Houssineau <jeremie.houssineau@ntu.edu.sg>.

*Proceedings of the 43rd International Conference on Machine Learning*, Seoul, South Korea. PMLR 306, 2026. Copyright 2026 by the author(s).

are realized using different distributional families, including Dirichlet distributions over class probabilities (Sensoy et al., 2018; Malinin & Gales, 2018; Charpentier et al., 2020), exponential-family predictive models (Charpentier et al., 2022), and Gaussian distributions over logits (Ghosh et al., 2016; Tagasovska & Lopez-Paz, 2019). Despite their efficiency, these approaches adopt heuristic objectives without rigorous justification for uncertainty quantification.

These limitations reveal a dilemma in epistemic uncertainty modeling: Bayesian methods offer theoretical rigor but remain intractable, while second-order predictors are efficient but lack principled connections between their objectives and epistemic uncertainty. To resolve this dilemma, we examine its foundation: both approaches are grounded in probability theory, which enforces complete allocation of probability mass and thus presumes sufficient knowledge is available to specify a precise distribution. Such a presumption, however, conflicts with epistemic uncertainty arising from incomplete knowledge. To address this conflict, we turn to possibility theory (Zadeh, 1978; Dubois & Prade, 1988), an alternative framework for epistemic uncertainty. It replaces probabilistic additivity and sum-normalization with supremum-based operators and max-normalization, naturally modeling epistemic uncertainty under incomplete knowledge (Dubois & Prade, 2021; 2024; Thomas & Houssineau, 2024). Despite the natural fit and its introduction decades ago, possibility theory remains largely unexplored in deep learning.

To bridge this gap, we introduce **D**irichlet-**a**pproximated **p**ossibilistic **p**osterior **p**redictions (DAPPr), a principled framework leveraging possibility theory for second-order predictors, achieving theoretical rigor and computational efficiency. DAPPr defines a possibilistic posterior over parameters, projects it to the prediction space via supremum operators to obtain a principled epistemic uncertainty function, and approximates it using learned Dirichlet possibility functions. This projection-and-approximation strategy yields a closed-form solution under the high-capacity assumption inherent to modern overparameterised networks and the widely used cross-entropy loss, resulting in a simple training objective with minimal regularisation (Figure 1). Despite the simplicity, extensive experiments show that DAPPr achieves superior uncertainty quantification performance over state-of-the-art second-order predictors across diverse benchmarks, including standard classification, long-tailed distributions, distribution shift detection, fine-grained classification tasks and uncertainty quantification on LLMs.

Our contributions are as follows:

i. We introduce DAPPr, a principled framework grounding second-order predictors in possibility theory for epistemic uncertainty modelling.

ii. We derive a tractable closed-form solution under the high-capacity assumption and the cross-entropy loss,

Pytorch-style pseudo code for DAPPr loss

```python
import torch.nn.functional as F
def DAPPr_loss(logits, labels, lamb, eps=1e-8):
  alpha = F.softplus(logits) + 1
  y = F.one_hot(labels, logits.shape[-1]).float()
  a_star = alpha - y + eps # avoid log instability
  p_star=(a_star / a_star.sum(dim=1,keepdim=True)).detach()
  a_0 = alpha.sum(1)
  loss=a_0*a_0.log()+(alpha*(p_star/alpha).log()).sum(dim=1)
  reg = (alpha * (1 - y)).square().sum(dim=1)
  return loss.mean() + lamb * reg.mean()
for x, labels in train_loader:  # Training Loop
  logits = model(x)
- loss = F.cross_entropy(logits, labels)
+ loss = DAPPr_loss(logits, labels, lamb)
  ...
```

*Figure 1.* PyTorch-style pseudocode for the DAPPr loss (∼10 lines of code) and its usage by simply replacing the cross-entropy loss.

achieving efficiency while maintaining rigour.

iii. We demonstrate competitive performance over state-of-the-art methods across diverse benchmarks.

## 2. Related work

**Bayesian Deep Learning.** Bayesian methods provide a principled framework for epistemic uncertainty by quantifying disagreement among plausible parameters through marginalization over the posterior distribution of parameters (Neal, 2012). However, exact marginalization is computationally intractable for modern networks (Wilson & Izmailov, 2020), necessitating approximations such as variational inference (Blundell et al., 2015; Graves, 2011), Monte Carlo dropout (MC Drop) (Gal & Ghahramani, 2016; Ovadia et al., 2019; Mobiny et al., 2019), and deep ensembles (Lakshminarayanan et al., 2017; Wen et al., 2020; Löhr et al., 2025). While effective, these methods rely on multiple models or extensive sampling, making them expensive and challenging to scale (Liu et al., 2020; He et al., 2020; Mukhoti et al., 2023). Despite recent efficiency efforts (Wilson & Izmailov, 2020; Rudner et al., 2022), the tension between theoretical rigor and computational tractability persists.

**Second-Order Predictive Modeling.** An alternative paradigm bypasses parameter-space inference by modeling distributions over predictions. These methods learn predictive distributions whose properties, such as concentration or dispersion, serve as proxies for epistemic uncertainty. Representative approaches include prior networks trained with KL divergence (KL-PN) (Malinin & Gales, 2018) and reverse KL divergence (RKL-PN) (Malinin & Gales, 2019), DUQ (Van Amersfoort et al., 2020) which estimates uncertainty via feature distances, PostNet (Charpentier et al., 2020) which employ normalizing flows for Dirichlet modeling, Natural Posterior Networks (Charpentier et al., 2022) which extend to exponential families, and RS-NN (Manchingal et al., 2025) which predicts set-valued outputs. Despite their efficiency, these methods lack principled foundations, relying on heuristic interpretations of epistemic uncertainty.

**Evidential Deep Learning.** Evidential deep learning (EDL)

(Sensoy et al., 2018) has emerged as a leading second-order approach by interpreting network outputs as Dirichlet distribution parameters over class probabilities. Deriving its objective from subjective logic (Jøsang, 2016) and Dempster-Shafer theory (Dempster, 1968; Shafer, 1976), EDL models both aleatoric and epistemic uncertainty through the Dirichlet's total evidence (concentration parameter), with higher evidence indicating lower epistemic uncertainty. However, this connection lacks rigorous justification (Bengs et al., 2022), and EDL can exhibit pathological behaviors such as increasing uncertainty with more data (Bengs et al., 2022; Yoon & Kim, 2025). Subsequent variants address specific issues: $\mathcal{I}$-EDL (Deng et al., 2023) introduces Fisher information regularisation, R-EDL (Chen et al., 2024) relaxes training constraints, DA-EDL (Yoon & Kim, 2024) incorporates density-aware mechanisms, and $\mathcal{F}$-EDL (Yoon & Kim, 2025) employs flexible Dirichlet modeling. Despite these improvements, EDL and its variants fundamentally lack principled probabilistic derivations rigorously connecting their objectives to epistemic uncertainty quantification.

**Possibility Theory.** Possibility theory (Zadeh, 1978; Dubois & Prade, 1988) provides an alternative framework for epistemic uncertainty from insufficient knowledge. Unlike probability theory's additive measures summing to one, possibility theory uses supremum-based operators normalizing to a maximum of one (Dubois, 2006), naturally accommodating imprecise information (Dubois & Prade, 2015). Possibilistic inference uses max-based rules rather than integration, offering computational advantages (Gebhardt & Kruse, 1995). Applications include point processes (Houssineau, 2021), control problems (Chen et al., 2021), and deep ensembles (Löhr et al., 2025). Recently, (Hieu et al., 2025) established a possibilistic foundation for decoupling epistemic and aleatoric uncertainty, highlighting the principled appeal of possibility theory in uncertainty modeling. Despite this appeal, possibility theory remains underexplored in deep learning for second-order predictors. Building on this foundation, we derive a tractable approach via possibilistic posterior projection and Dirichlet approximation, yielding a practical training objective for second-order predictors.

## 3. Preliminaries for Possibility Theory

Probability theory models uncertainty through additive measures normalized to one ($\int p(x)dx = 1$), thereby enforcing a complete allocation of probability mass. This complete allocation is appropriate to modeling aleatoric uncertainty from intrinsic data randomness, provided sufficient knowledge is available. However, when information is limited, complete probability mass allocation becomes ill-suited for representing epistemic uncertainty arising from ignorance.

The epistemic uncertainty, induced by a lack of knowledge, concerns which hypotheses cannot be excluded given limited information. This emphasis on non-exclusion is naturally formalised in *possibility theory* (Zadeh, 1978; Dubois, 2006), which represents uncertainty through plausibility measures and, at a fundamental level, decouples epistemic uncertainty from aleatoric uncertainty (Hieu et al., 2025).

In this section, we introduce key concepts from possibility theory. First, we define possibility functions, with Dirichlet possibility functions as an important special case. We then present essential operations: possibilistic Bayesian inference, change of variables, and divergence measures.

**Possibility function.** In possibility theory, uncertainty about a parameter $\theta \in \Theta$ is represented by a possibility function $f : \Theta \to [0, 1]$ satisfying $\sup_{\theta \in \Theta} f(\theta) = 1$. Given this representation, the possibility of an event $B \subseteq \Theta$ is defined as $\Pi(B) \doteq \sup_{\theta \in B} f(\theta)$. Accordingly, an event is impossible if $\Pi(B) = 0$, while $\Pi(B) = 1$ indicates that the event cannot be excluded given the available information. When such non-exclusion holds simultaneously for an event $B$ and its complement $B^c \doteq \Theta \setminus B$, *i.e.* $\Pi(B) = 1$ and $\Pi(B^c) = 1$, the representation provides no information to exclude either event. When this non-exclusion holds for all events, the representation corresponds to total ignorance, where $f(\theta) = 1$ for all $\theta \in \Theta$ and the possibility function is denoted by $\mathbf{1}$.

**Dirichlet possibility function.** When the parameter space $\Theta$ is a probability simplex $\Delta^{K-1} = \{\boldsymbol{p} \in [0,1]^K : \sum_{k=1}^K p_k = 1\}$, a natural structured possibility function on this simplex is given by the Dirichlet possibility function $\overline{\text{Dir}}(\cdot; \boldsymbol{\alpha})$ with $\boldsymbol{\alpha} \in [0, \infty)^K$ and $\alpha_0 = \sum_{k=1}^K \alpha_k$, defined as:

$$\overline{\text{Dir}}(\boldsymbol{p}; \boldsymbol{\alpha}) = \alpha_0^{\alpha_0} \prod_{k=1}^K \left(\frac{p_k}{\alpha_k}\right)^{\alpha_k}, \qquad (1)$$

with the convention that $\alpha_k^{\alpha_k} = 1$ when $\alpha_k = 0$. This formulation is both a conjugate prior for the multinomial likelihood, mirroring the Dirichlet distribution, and a valid possibility function satisfying $\sup_{\boldsymbol{p}} \overline{\text{Dir}}(\boldsymbol{p}; \boldsymbol{\alpha}) = 1$. The supremum is uniquely attained at the mode $\boldsymbol{p} = \boldsymbol{\alpha}/\alpha_0$ when $\alpha_0 > 0$, while spanning the entire simplex when $\alpha_0 = 0$ to represent total ignorance $\overline{\text{Dir}}(\cdot; \boldsymbol{\alpha}) = \mathbf{1}$.

**Possibilistic Bayes Posterior.** Having introduced possibility functions for representing epistemic uncertainty, we now introduce the general form of the Bayesian posterior in a possibilistic setting. Let the relationship between a parameter $\theta$ and a data point $y$ be characterised by a loss $\ell(\theta, y)$, and let prior on $\theta$ be represented by a possibility function $f(\theta)$. The corresponding possibilistic posterior is defined as

$$f(\theta \,|\, y) = \frac{\exp(-\ell(\theta, y))f(\theta)}{\sup_{\theta' \in \Theta} \exp(-\ell(\theta', y))f(\theta')}. \qquad (2)$$

Importantly, if the loss admits a probabilistic interpretation, *i.e.*, $\ell(\theta, y) = -\log p(y \,|\, \theta)$ for some distribution $p(\cdot \,|\, \theta)$, the posterior $f(\cdot|y)$ is consistent with the standard Bayesian

posterior and inherits familiar properties such as conjugacy and Bernstein–von Mises behaviour (Hieu et al., 2025).

**Possibilistic change of variable.** In inference, we often need to characterize uncertainty about a transformed quantity $\psi = T(\theta)$ rather than the original parameter $\theta$. Characterizing this uncertainty requires propagating uncertainty from $\theta$ through the mapping $T$, ensuring the plausibility of $\psi$ reflects the plausibility of $\theta$ that generates it. In probability theory, such propagation uses a Jacobian determinant since densities change under transformation. However, possibility functions are *not* densities. Therefore, possibilistic transformation cannot simply replace integration with supremum while retaining the Jacobian determinant. Instead, (Baudrit et al., 2008) define the possibilistic change of variable as:

$$f(\psi) = \sup_{\theta \in T^{-1}(\psi)} f(\theta), \qquad (3)$$

where $T^{-1}(\psi) \subseteq \Theta$ is the set-valued pre-image of $\psi$ (with $\sup_{\theta \in \emptyset} f(\theta) = 0$). This formulation assigns to $\psi$ the highest plausibility among all its pre-images.

**Maxitive pseudo-divergence.** Building on the concepts above, we now explain how to compare possibility functions. The comparison is based on a natural partial order: for two possibility functions $f$ and $g$ on $\Theta$, we write $f \preceq g$ if $f(\theta) \leq g(\theta)$ for all $\theta \in \Theta$. Under this relation, every possibility function satisfies $f \preceq \mathbf{1}$. This ordering further enables a pointwise comparison of possibility functions via the maxitive pseudo-divergence (Singh et al., 2025):

$$D_{\max}(f \| g) = \max_{\theta \in \Theta} \log \frac{f(\theta)}{g(\theta)} \geq 0. \qquad (4)$$

The non-negativity of $D_{\max}$ follows from the fact that both $f$ and $g$ have a supremum equal to 1, so that $f(\theta) < g(\theta)$ cannot hold everywhere on $\Theta$, and $f \preceq g$ requires that $f(\theta) = g(\theta)$ when $\theta$ is in $\arg\max_\theta f(\theta)$. It follows that $D_{\max}(f \| g) = 0$ when $f \preceq g$, rather than only when $f = g$.

## 4. Methodology

### 4.1. Motivation

A standard neural network trained for classification maps an input $\boldsymbol{x} \in \mathcal{X}$ to a probability vector $\boldsymbol{p} \in \Delta^{K-1}$ over $K$ classes. This mapping is realised by a predictive model $\Phi_{\boldsymbol{\theta}} : \mathcal{X} \to \Delta^{K-1}$, parametrised by $\boldsymbol{\theta} \in \Theta$ and trained on a dataset $\mathcal{D} \subseteq \{\mathcal{X} \times \mathcal{Y}\}$ by minimizing an empirical risk:

$$L(\boldsymbol{\theta}; \mathcal{D}) = \sum_{(\boldsymbol{x}, \boldsymbol{y}) \in \mathcal{D}} \ell(\Phi_{\boldsymbol{\theta}}(\boldsymbol{x}), \boldsymbol{y}), \qquad (5)$$

where $\ell(\cdot, \cdot)$ is a standard classification loss, such as the widely-adopted cross-entropy $\ell(\boldsymbol{p}, \boldsymbol{y}) = -\sum_{k=1}^{K} y_k \log p_k$. After training, the learned $\boldsymbol{\theta}$ produces a point prediction $\boldsymbol{p}_{\text{test}} = \Phi_{\boldsymbol{\theta}}(\boldsymbol{x}_{\text{test}})$ for a test input $\boldsymbol{x}_{\text{test}}$. While this point prediction provides class probabilities, it does not capture

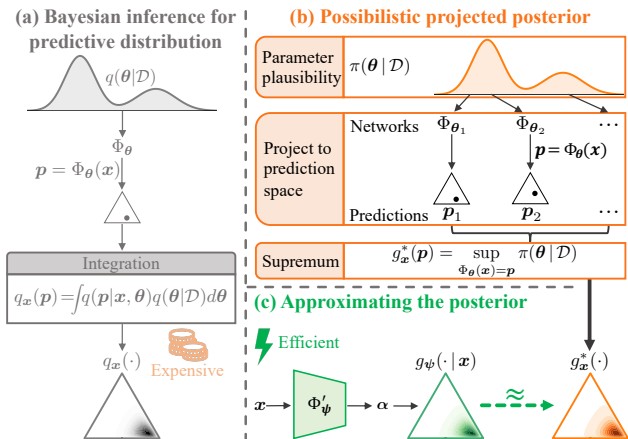

*Figure 2.* Overview of DAPPr. (a) Bayesian inference obtains a predictive distribution $q_{\boldsymbol{x}}(\cdot)$ by marginalising over parameters, but is computationally intractable. (b) DAPPr instead operates under possibility theory, projecting parameter plausibility to prediction space to define a principled projected posterior $g_{\boldsymbol{x}}^*(\cdot)$ as a reference. (c) A network $\Phi_{\boldsymbol{\psi}}'$ then learns a Dirichlet possibility function $g_{\boldsymbol{\psi}}(\cdot | \boldsymbol{x})$ to efficiently approximate the reference $g_{\boldsymbol{x}}^*(\cdot)$.

epistemic uncertainty and thus cannot indicate prediction reliability, especially when $\boldsymbol{x}_{\text{test}}$ is weakly supported by $\mathcal{D}$.

To model epistemic uncertainty, we seek a prediction-level uncertainty function that assigns plausibility to each prediction $\boldsymbol{p}$ for an input $\boldsymbol{x}$. Such a function can be constructed in principle via Bayesian learning, which projects the parameter posterior $q(\boldsymbol{\theta}|\mathcal{D})$ to prediction space by integrating over all parameters, yielding a predictive distribution $q_{\boldsymbol{x}}(\boldsymbol{p}) = \int q(\boldsymbol{p}|\boldsymbol{x}, \boldsymbol{\theta}) q(\boldsymbol{\theta}|\mathcal{D}) d\boldsymbol{\theta}$ quantifying plausibility (Fig. 2 (a)). However, performing this over the high-dimensional parameter space of deep networks is intractable.

A more efficient alternative is a second-order predictor directly mapping each $\boldsymbol{x}$ to a distribution over predictions $\boldsymbol{p}$ quantifying their plausibility. Since predictions in classification lie on the probability simplex, the Dirichlet possibility function in Eq. (1) naturally represents such a distribution. We thus instantiate the predictor as a network $\Phi_{\boldsymbol{\psi}}'$ parameterised by $\boldsymbol{\psi}$ that maps $\boldsymbol{x}$ to Dirichlet parameters $\boldsymbol{\alpha}$ defining a Dirichlet possibility function $g_{\boldsymbol{\psi}}(\cdot | \boldsymbol{x})$ that represents plausibility over predictions to capture epistemic uncertainty.

Ideally, if Dirichlet parameters $\boldsymbol{\alpha}$ or uncertainty labels were available for each input $\boldsymbol{x}$, training would be straightforward. However, such supervision is unavailable in practice. Without explicit supervision, we instead train $g_{\boldsymbol{\psi}}(\cdot | \boldsymbol{x})$ by minimising its discrepancy with a reference function that captures epistemic uncertainty. In probability theory, a natural such reference is the predictive distribution $q_{\boldsymbol{x}}(\cdot)$ (Figure 2 (a)), but obtaining this is intractable in modern deep networks. We therefore turn to possibility theory, which resolves this intractability while maintaining rigour. As illustrated in Figure 2 (b) & (c), we construct a well-defined reference function, the possibilistic projected posterior $g_{\boldsymbol{x}}^*(\cdot)$

and train $g_{\psi}(\cdot|\boldsymbol{x})$ to approximate it efficiently.

## 4.2. Defining the objective

To construct a principled reference function, we examine where epistemic uncertainty about a prediction comes from. The prediction for an input $\boldsymbol{x}$ is deterministically given by $\boldsymbol{p} = \Phi_{\boldsymbol{\theta}}(\boldsymbol{x})$: once $\boldsymbol{\theta}$ is known, the prediction follows without ambiguity. This means epistemic uncertainty about a prediction can only arise from uncertainty about the parameters that produced it. In other words, how much we trust a prediction comes from how plausible the model is.

The plausibility of a model parameter $\boldsymbol{\theta}$ thus governs epistemic uncertainty about predictions, which we quantify by relating it to the empirical loss $L(\boldsymbol{\theta}; \mathcal{D})$ defined in Eq. (5). The intuition is simple: parameters with lower loss better fit the observed data and are thus more plausible. This intuition can be formalised using the possibilistic posterior in Eq. (2). Under a uniform prior possibility function $\mathbf{1}$, which assumes no prior preference for any particular parameters, the possibilistic Bayesian posterior becomes:

$$\pi(\boldsymbol{\theta}\,|\,\mathcal{D}) = \frac{\exp(-L(\boldsymbol{\theta}; \mathcal{D}))}{\sup_{\boldsymbol{\theta}' \in \Theta} \exp(-L(\boldsymbol{\theta}'; \mathcal{D}))}. \quad (6)$$

This posterior assigns higher plausibility to parameters with lower empirical loss. The normalisation by the supremum ensures $\sup_{\boldsymbol{\theta}} \pi(\boldsymbol{\theta}\,|\,\mathcal{D}) = 1$, satisfying the requirement of a valid possibility function. Particularly, when the loss is cross-entropy loss, the resulting posterior reduces to the relative likelihood, a well-established prior-free approach in statistical inference (Birnbaum, 1962; Wasserman, 1990; Walley & Moral, 1999; Giang & Shenoy, 2012), with axiomatic justification based on the likelihood principle, compatibility with Bayes' rule, and the minimal commitment principle (Denoeux, 2014; Löhr et al., 2025).

With the posterior $\pi(\cdot\,|\,\mathcal{D})$ quantifying plausibility over parameters, the next step is transferring it to the prediction space to determine which predictions are plausible. For a given input $\boldsymbol{x}$, we project the posterior onto the simplex $\Delta^{K-1}$ via the mapping $\boldsymbol{\theta} \mapsto \Phi_{\boldsymbol{\theta}}(\boldsymbol{x})$. By the possibilistic change-of-variable rule in Eq. (3), the projection yields:

$$g_{\boldsymbol{x}}^*(\boldsymbol{p}) = \sup\{\pi(\boldsymbol{\theta}\,|\,\mathcal{D}) : \boldsymbol{\theta} \in \Theta, \, \Phi_{\boldsymbol{\theta}}(\boldsymbol{x}) = \boldsymbol{p}\}. \quad (7)$$

The function assigns to each prediction $\boldsymbol{p}$ the maximum plausibility among all parameters that yields $\boldsymbol{p}$ for input $\boldsymbol{x}$. We denote this assignment compactly by $\sup_{\Phi_{\boldsymbol{\theta}}(\boldsymbol{x})=\boldsymbol{p}} \pi(\boldsymbol{\theta}\,|\,\mathcal{D})$.

This supremum-based projection plays a similar role to marginalisation in Bayesian inference, where the predictive distribution $q_{\boldsymbol{x}}(\cdot)$ is obtained by integrating the parameter posterior over the high-dimensional $\Theta$. Such marginalisation is generally intractable, requiring approximations such as ensembles. In contrast, Eq. (7) replaces integration with constrained optimisation. While non-trivial, this admits

tractable approximations without ensembles.

We realise this approximation by learning a second-order predictor $\Phi'_{\boldsymbol{\psi}}$ whose output is a possibility function $g_{\boldsymbol{\psi}}(\cdot|\boldsymbol{x})$ over $\boldsymbol{p}$. This output is matched to the projected posterior $g_{\boldsymbol{x}}^*(\cdot)$ by minimising the pseudo-divergence $D_{\max}$ in (4):

$$\mathcal{L}(\boldsymbol{\psi}; \mathcal{D}) = \mathbb{E}_{\boldsymbol{x} \sim \mathcal{X}}\left[D_{\max}\left(g_{\boldsymbol{\psi}}(\cdot\,|\,\boldsymbol{x}) \,\|\, g_{\boldsymbol{x}}^*(\cdot)\right)\right]$$

$$= \mathbb{E}_{\boldsymbol{x} \sim \mathcal{X}}\left[\max_{\boldsymbol{p} \in \Delta}\left(\log g_{\boldsymbol{\psi}}(\boldsymbol{p}\,|\,\boldsymbol{x}) - \log g_{\boldsymbol{x}}^*(\boldsymbol{p})\right)\right]. \quad (8)$$

This objective enforces the ordering $g_{\boldsymbol{\psi}}(\cdot\,|\,\boldsymbol{x}) \preceq g_{\boldsymbol{x}}^*(\cdot)$ by penalising the maximal pointwise ratio between the learned possibility function and the projected posterior. With an uninformative initialisation of $g_{\boldsymbol{\psi}}$, optimisation reduces $g_{\boldsymbol{\psi}}$ only where this ordering is violated and once satisfied, the divergence vanishes and cannot be further reduced (Sec. 3).

Minimising the loss $\mathcal{L}(\boldsymbol{\psi}; \mathcal{D})$ in Eq. (8) yields a min-max problem, with an inner maximisation over $\boldsymbol{p} \in \Delta$ and an outer minimisation over $\boldsymbol{\psi}$. Directly optimising it is challenging because the inner maximiser depends on $\boldsymbol{\psi}$. To address this, we apply Danskin's theorem (Danskin, 1967), which computes the gradient of the outer objective by differentiating at the inner maximum, assuming the maximiser is unique; otherwise, the update is interpreted as a subgradient. Accordingly, the inner maximizer $\boldsymbol{p}^*$ for an input $\boldsymbol{x}$ is:

$$\boldsymbol{p}^* = \arg\max_{\boldsymbol{p} \in \Delta}\left[\log g_{\boldsymbol{\psi}}(\boldsymbol{p}\,|\,\boldsymbol{x}) - \log g_{\boldsymbol{x}}^*(\boldsymbol{p})\right]. \quad (9)$$

This shows that $\boldsymbol{p}^*$ is the point at which $g_{\boldsymbol{\psi}}(\cdot\,|\,\boldsymbol{x})$ most overestimates $g_{\boldsymbol{x}}^*(\cdot)$, i.e., where their ratio is maximal. With the inner maximiser $\boldsymbol{p}^*$ obtained at each iteration, we can learn the second-order predictor by minimising $\mathcal{L}$ via stochastic gradient descent, pushing $\boldsymbol{\psi}$ to decrease $g_{\boldsymbol{\psi}}$ at this point.

## 4.3. Approximating the objective

Learning the second-order predictor would be straightforward if the conceptually well-defined $g_{\boldsymbol{x}}^*(\cdot)$ in Eq. (9) were available. However, evaluating this projected posterior involves constrained optimisation over the high-dimensional parameter space $\Theta$, which renders direct computation intractable. To overcome this intractability, we derive in this subsection a tractable surrogate for $g_{\boldsymbol{x}}^*$ that enables efficient optimisation of the inner maximisation in Eq. (9).

For a training sample $(\boldsymbol{x}, \boldsymbol{y}) \in \mathcal{D}$, the negative log-possibility $-\log g_{\boldsymbol{x}}^*(\boldsymbol{p})$ in Eq. (9) can be expressed as

$$-\log g_{\boldsymbol{x}}^*(\boldsymbol{p}) = -\log \sup_{\Phi_{\boldsymbol{\theta}}(\boldsymbol{x})=\boldsymbol{p}} \pi(\boldsymbol{\theta}\,|\,\mathcal{D})$$

$$= \ell(\boldsymbol{p}, \boldsymbol{y}) + \inf_{\Phi_{\boldsymbol{\theta}}(\boldsymbol{x})=\boldsymbol{p}} L(\boldsymbol{\theta}; \mathcal{D} \setminus \{(\boldsymbol{x}, \boldsymbol{y})\}) + c. \quad (10)$$

Here $c = \log \sup_{\boldsymbol{\theta}'} \exp(-L(\boldsymbol{\theta}'; \mathcal{D}))$ is a constant independent of $\boldsymbol{p}$ and $\boldsymbol{x}$, and therefore can be safely ignored.

Once $c$ is ignored, evaluating Eq. (10) reduces to analysing how the empirical risk behaves under the con-

straint $\Phi_{\boldsymbol{\theta}}(\boldsymbol{x}) = \boldsymbol{p}$. This constraint fixes the contribution of the sample $(\boldsymbol{x}, \boldsymbol{y})$ to the empirical risk $L(\boldsymbol{\theta}; \mathcal{D})$ as $\ell(\boldsymbol{p}, \boldsymbol{y})$, reducing the problem to the infimum over the leave-one-out loss $L(\boldsymbol{\theta}; \mathcal{D} \setminus \{(\boldsymbol{x}, \boldsymbol{y})\})$.

The key observation is that this infimum depends on $\boldsymbol{p}$ only through the constraint $\Phi_{\boldsymbol{\theta}}(\boldsymbol{x}) = \boldsymbol{p}$. Although this constraint couples all samples through the shared parameters $\boldsymbol{\theta}$, a standard assumption in the overparameterised regime is that modern deep networks have sufficient capacity to fit any individual sample without substantially affecting predictions on other samples (Hornik et al., 1989; Zhang et al., 2017; Koh & Liang, 2017). Under this sufficient capacity condition, the constraint can be satisfied for any $\boldsymbol{p}$ without affecting the infimum of the leave-one-out loss. This infimum is thus approximately a constant independent of $\boldsymbol{p}$:

$$\inf_{\Phi_{\boldsymbol{\theta}}(\boldsymbol{x}) = \boldsymbol{p}} L(\boldsymbol{\theta}; \mathcal{D} \setminus \{(\boldsymbol{x}, \boldsymbol{y})\}) \approx c_{\boldsymbol{x}}, \qquad (11)$$

where $c_{\boldsymbol{x}}$ is a constant independent of $\boldsymbol{p}$. We empirically verify this approximation in §D. Under this approximation, the projected posterior simplifies to the tractable surrogate:

$$g_{\boldsymbol{x}}^*(\boldsymbol{p}) \propto \exp\big(-\ell(\boldsymbol{p}, \boldsymbol{y})\big), \qquad (12)$$

yielding a practical approximation of $\boldsymbol{p}^*$ in Eq. (9) as:

$$\tilde{\boldsymbol{p}}^* = \arg\max_{\boldsymbol{p} \in \Delta} \log g_{\boldsymbol{\psi}}(\boldsymbol{p} \mid \boldsymbol{x}) + \ell(\boldsymbol{p}, \boldsymbol{y}). \qquad (13)$$

We note that the approximation in Eq. (11) may be violated in the rare case where samples share the same $\boldsymbol{x}$ but have conflicting labels $\boldsymbol{y}, \boldsymbol{y}'$. Here, the constraint induces coupled effects, and a faithful surrogate replaces Eq. (12) with:

$$g_{\boldsymbol{x}}^*(\boldsymbol{p}) \propto \exp\big(-\ell(\boldsymbol{p}, \boldsymbol{y}) - \ell(\boldsymbol{p}, \boldsymbol{y}')\big),$$

reflecting the interaction induced by $\Phi_{\boldsymbol{\theta}}$ at $\boldsymbol{x}$.

Beyond this specific case, the approximation in Eq. (11) is most appropriate for modern overparameterised networks, such as ResNets (He et al., 2016) and Transformers (Vaswani et al., 2017) where the sufficient capacity assumption naturally holds; in low-capacity settings where the model struggles to fit the data, uncertainty quantification is of secondary concern and the approximation is less relevant.

### 4.4. Parametrising the Dirichlet possibility function

With the approximate maximiser $\tilde{\boldsymbol{p}}^*$ in Eq. (13) established, training the second-order predictor $\Phi_{\boldsymbol{\psi}}'$ requires evaluating the surrogate objective associated with the base prediction loss $\ell$. In classification, this base loss is standardly taken to be the cross-entropy, which we therefore adopt in the sequel. Under the cross-entropy loss, evaluating the surrogate objective in Eq. (13) necessitates an explicit functional form for the learned possibility function $g_{\boldsymbol{\psi}}$. We therefore realise $g_{\boldsymbol{\psi}}(\cdot \mid \boldsymbol{x})$ as a Dirichlet possibility function whose

parameters are produced by $\Phi_{\boldsymbol{\psi}}'$. According to Eq. (1), the corresponding log-possibility for a given input $\boldsymbol{x}$ and $\boldsymbol{p}$ is:

$$\log g_{\boldsymbol{\psi}}(\boldsymbol{p} \mid \boldsymbol{x}) = \alpha_0 \log \alpha_0 + \sum_{k=1}^{K} \alpha_k \log \frac{p_k}{\alpha_k}, \qquad (14)$$

where $\boldsymbol{\alpha} = \Phi_{\boldsymbol{\psi}}'(\boldsymbol{x})$ and $\alpha_0 = \|\boldsymbol{\alpha}\|_1 = \sum_{k=1}^{K} \alpha_k$. Under this parametrisation, the surrogate maximisation in Eq. (13) admits a closed-form solution.

**Proposition 1.** *Let the loss be the cross-entropy loss and let $g_{\boldsymbol{\psi}}$ be parameterised as a Dirichlet possibility function with $\boldsymbol{\alpha} = \Phi_{\boldsymbol{\psi}}'(\boldsymbol{x})$. Then, for a labelled sample $(\boldsymbol{x}, \boldsymbol{y})$ with one-hot label vector $\boldsymbol{y}$, the approximate maximiser is*

$$\tilde{\boldsymbol{p}}^* = \frac{1}{\alpha_0 - 1}(\boldsymbol{\alpha} - \boldsymbol{y}). \qquad (15)$$

Prop. 1 (proof in §A) shows that, under a Dirichlet parametrisation and cross-entropy loss, the surrogate maximiser $\tilde{\boldsymbol{p}}^*$ admits closed-form expression. This expression is valid only if $\tilde{\boldsymbol{p}}^*$ lies in the probability simplex, which requires $\alpha_k > 1$ for all $k$. We thus implement $\boldsymbol{\alpha}$ using softplus$(\cdot) + 1$ parameterization to satisfy this requirement. Although this implementation precludes the exact uninformative initialisation discussed in Sec. 4.2, standard random network initialisation is sufficient for training to proceed. At initialization, it produces nearly equal $\alpha_k$ across classes, causing $g_{\boldsymbol{\psi}}(\cdot | \boldsymbol{x})$ to concentrate around the uniform prediction. Since $g_{\boldsymbol{x}}^*$ is concentrated around the true label and $g_{\boldsymbol{\psi}}$ is max-normalised by construction, $g_{\boldsymbol{\psi}}(\cdot | \boldsymbol{x})$ initially over-estimates $g_{\boldsymbol{x}}^*$ at its mode, ensuring a meaningful overestimation point and progressively driving the mode of $g_{\boldsymbol{\psi}}(\cdot | \boldsymbol{x})$ toward that of $g_{\boldsymbol{x}}^*$.

Substituting $\tilde{\boldsymbol{p}}^*$ into Eq. (14) yields a per-sample surrogate likelihood $\log g_{\boldsymbol{\psi}}(\tilde{\boldsymbol{p}}^* \mid \boldsymbol{x})$, enabling independent per-sample fitting but potentially driving $g_{\boldsymbol{\psi}}(\cdot \mid \boldsymbol{x})$ to become arbitrarily sharp, corresponding to large total evidence $\alpha_0$ around the true label. To prevent this, we penalise evidence on incorrect classes via a spurious evidence regulariser:

$$\mathcal{R}(\boldsymbol{x}) = \|(\mathbf{1} - \boldsymbol{y}) \odot \boldsymbol{\alpha}\|_2^2. \qquad (16)$$

This regulariser enforces the minimum commitment principle of possibility theory, which prescribes that plausibility should not exceed what the evidence supports. By suppressing evidence on incorrect classes unsupported by the observed label, the regulariser avoids assigning plausibility contradicted by the label, acting as a structured constraint on wrong-class evidence that improves discrimination.

Combining the surrogate likelihood $\log g_{\boldsymbol{\psi}}(\tilde{\boldsymbol{p}}^* \mid \boldsymbol{x})$ with this regulariser $\mathcal{R}(\boldsymbol{x})$ yields the final per-sample training loss:

$$\ell_{\boldsymbol{\psi}}(\boldsymbol{x}) = \log g_{\boldsymbol{\psi}}(\tilde{\boldsymbol{p}}^* \mid \boldsymbol{x}) + \lambda \mathcal{R}(\boldsymbol{x}), \qquad (17)$$

where $\lambda \geq 0$ controls regularisation strength. The resulting pipeline in Figure 3 is implemented by simply replacing the cross-entropy loss with Eq. (17) during training (Figure 1).

*Table 1.* AUPR (↑) on CIFAR-10/100. Conf.: confidence estimation via aleatoric uncertainty. OOD detection using epistemic uncertainty against SVHN and CIFAR-100 (for CIFAR-10) or TinyImageNet (for CIFAR-100). EDL-based results are from (Yoon & Kim, 2025).

| Method | CIFAR-10 | | | | CIFAR-100 | | | |
|---|---|---|---|---|---|---|---|---|
| | Test Acc. | Conf. | OOD Detection | | Test Acc. | Conf. | OOD Detection | |
| | | | SVHN | CIFAR-100 | | | SVHN | TinyImageNet |
| MC Drop | $82.84_{\pm0.1}$ | $97.15_{\pm0.0}$ | $51.39_{\pm0.1}$ | $45.57_{\pm1.0}$ | $65.94_{\pm0.6}$ | $92.00_{\pm0.3}$ | $71.83_{\pm2.0}$ | $74.93_{\pm0.6}$ |
| DUQ | $88.79_{\pm1.5}$ | $98.35_{\pm0.4}$ | $86.62_{\pm3.1}$ | $85.28_{\pm0.6}$ | $57.93_{\pm0.9}$ | $84.30_{\pm0.9}$ | $64.90_{\pm5.8}$ | $71.64_{\pm0.5}$ |
| KL-PN | $23.94_{\pm2.3}$ | $35.22_{\pm7.5}$ | $49.56_{\pm3.6}$ | $58.18_{\pm4.5}$ | $41.72_{\pm0.6}$ | $82.42_{\pm0.7}$ | $47.03_{\pm2.9}$ | $67.33_{\pm0.2}$ |
| RKL-PN | $85.48_{\pm1.6}$ | $97.12_{\pm0.6}$ | $65.36_{\pm2.2}$ | $69.37_{\pm1.4}$ | $69.92_{\pm0.9}$ | $93.12_{\pm0.3}$ | $67.92_{\pm3.5}$ | $74.60_{\pm0.1}$ |
| PostNet | $88.54_{\pm0.5}$ | $98.38_{\pm0.1}$ | $82.34_{\pm0.5}$ | $82.25_{\pm3.8}$ | $54.69_{\pm1.1}$ | $85.56_{\pm0.7}$ | $54.92_{\pm2.1}$ | $61.68_{\pm1.2}$ |
| NatPN | $90.29_{\pm0.1}$ | $98.58_{\pm0.4}$ | $87.12_{\pm1.1}$ | $85.51_{\pm1.3}$ | $65.95_{\pm0.9}$ | $90.84_{\pm0.8}$ | $68.98_{\pm0.4}$ | $74.94_{\pm0.8}$ |
| RSNN | $91.91_{\pm0.3}$ | $99.19_{\pm0.0}$ | $89.68_{\pm0.6}$ | $88.94_{\pm0.0}$ | $68.60_{\pm0.4}$ | $92.73_{\pm0.0}$ | $73.51_{\pm3.1}$ | $78.08_{\pm0.0}$ |
| EDL | $83.55_{\pm0.6}$ | $97.86_{\pm0.2}$ | $79.12_{\pm3.7}$ | $84.18_{\pm0.7}$ | $45.91_{\pm5.6}$ | $91.28_{\pm0.8}$ | $56.21_{\pm3.1}$ | $70.13_{\pm2.0}$ |
| $\mathcal{I}$-EDL | $89.20_{\pm0.3}$ | $98.72_{\pm0.1}$ | $82.96_{\pm2.2}$ | $84.84_{\pm0.6}$ | $66.38_{\pm0.5}$ | $92.84_{\pm0.1}$ | $67.51_{\pm2.9}$ | $75.86_{\pm0.3}$ |
| R-EDL | $90.09_{\pm0.3}$ | $98.98_{\pm0.1}$ | $85.00_{\pm1.2}$ | $87.73_{\pm0.3}$ | $63.53_{\pm0.5}$ | $92.69_{\pm0.2}$ | $61.80_{\pm3.4}$ | $69.78_{\pm1.3}$ |
| DAEDL | $91.11_{\pm0.2}$ | $99.08_{\pm0.0}$ | $85.54_{\pm1.4}$ | $88.19_{\pm0.1}$ | $66.01_{\pm2.6}$ | $86.00_{\pm0.3}$ | $72.07_{\pm4.1}$ | $77.40_{\pm1.6}$ |
| $\mathcal{F}$-EDL | $91.19_{\pm0.2}$ | $99.10_{\pm0.0}$ | $91.20_{\pm1.3}$ | $88.37_{\pm0.3}$ | $69.40_{\pm0.2}$ | $94.01_{\pm0.1}$ | $\mathbf{75.35_{\pm2.3}}$ | $\mathbf{80.58_{\pm0.2}}$ |
| DAPPr | $\mathbf{92.00_{\pm0.2}}$ | $\mathbf{99.23_{\pm0.0}}$ | $\mathbf{91.72_{\pm1.2}}$ | $\mathbf{89.39_{\pm0.3}}$ | $\mathbf{70.85_{\pm0.2}}$ | $\mathbf{94.39_{\pm0.1}}$ | $73.32_{\pm3.3}$ | $79.11_{\pm0.1}$ |
| Ensemble | $93.82_{\pm0.1}$ | $99.49_{\pm0.0}$ | $86.90_{\pm0.0}$ | $89.92_{\pm0.1}$ | $\mathbf{75.48_{\pm0.2}}$ | $94.89_{\pm0.1}$ | $\mathbf{75.09_{\pm0.2}}$ | $80.02_{\pm0.0}$ |
| +DAPPr | $93.63_{\pm0.0}$ | $\mathbf{99.51_{\pm0.0}}$ | $\mathbf{94.73_{\pm0.2}}$ | $\mathbf{91.45_{\pm0.1}}$ | $74.38_{\pm0.2}$ | $\mathbf{95.38_{\pm0.1}}$ | $74.31_{\pm0.4}$ | $\mathbf{80.88_{\pm0.1}}$ |

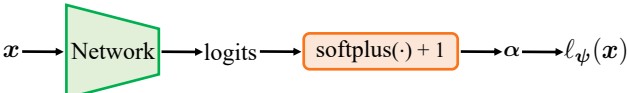

*Figure 3.* DAPPr pipeline. Logits are passed through softplus$(\cdot)+1$ to obtain Dirichlet parameters $\boldsymbol{\alpha}$ satisfying $\alpha_k > 1$ (required by Prop. 1), which are then used to minimise $\ell_\psi(\boldsymbol{x})$ in Eq. (17).

After training, $g_\psi(\cdot\,|\,\boldsymbol{x})$ has mode $\boldsymbol{\alpha}/\alpha_0$ with $\alpha_0$ controlling its concentration. At the mode, the most plausible prediction, only irreducible data ambiguity remains, so we measure aleatoric uncertainty as $1 - \max_k \alpha_k/\alpha_0$, which is small when one class dominates and large when classes compete. As $\alpha_0$ decreases, $g_\psi(\cdot\,|\,\boldsymbol{x})$ broadens and approaches total ignorance, so we measure epistemic uncertainty as $K/\alpha_0$, directly reflecting insufficient parameter knowledge. Despite sharing the Dirichlet parametrisation with EDL, DAPPr differs fundamentally, as discussed in §B.

# 5. Experiments

We conduct comprehensive experiments to evaluate our uncertainty modeling method across diverse settings and compare it with state-of-the-art second-order predictors.

## 5.1. Experimental Setup

**Datasets.** Following (Deng et al., 2023; Chen et al., 2024; Yoon & Kim, 2025), we evaluate on CIFAR-10 and CIFAR-100 (Krizhevsky et al., 2009). We further include CIFAR-10-LT (Cui et al., 2019), an artificially imbalanced version of CIFAR-10 with imbalance ratio $\rho$, two fine-grained datasets CUB-200-2011 (Wah et al., 2011) and Stanford-Dogs (Khosla et al., 2011) with numerous categories and high-resolution images and TinyImageNet (Deng et al., 2009) with more classes and images.

**Evaluation Metrics.** We assess classification accuracy, con-

fidence estimation, and out-of-distribution (OOD) detection. Following (Kendall & Gal, 2017; Hüllermeier & Waegeman, 2021), aleatoric uncertainty serves as a proxy for confidence estimation with confidence defined as negative aleatoric uncertainty and OOD scores as negative epistemic uncertainty. Both are evaluated by the area under the precision-recall curve (AUPR, normalized to 0-100, higher is better) with labels correct/ID=1 and incorrect/OOD=0. For OOD datasets, we use SVHN (Netzer et al., 2011) and CIFAR-100 for CIFAR-10; SVHN and TinyImageNet (Deng et al., 2009) for CIFAR-100, and ImageNet-O (Hendrycks et al., 2021), DTD (Cimpoi et al., 2014) and Places365 (Zhou et al., 2017) for fine-grained datasets and TinyImageNet.

**Implementation**. Following (Charpentier et al., 2020), we use VGG16 (Simonyan & Zisserman, 2014) for CIFAR-10/CIFAR-10-LT, ResNet-18 (He et al., 2016) for CIFAR-100, and ResNet-50 for fine-grained datasets and TinyImageNet. We train for up to 100 epochs (200 for fine-grained datasets) with batch size 64 (256 for fine-grained and TinyImageNet) using validation-based early stopping. We report mean ± std over 5 random seeds. See §C for details.

## 5.2. Comparison with state-of-the-art methods

**CIFAR-10/100.** Table 1 shows that our method achieves the best accuracy and confidence estimation on both CIFAR-10 and CIFAR-100, with competitive OOD detection. It outperforms $\mathcal{I}$-EDL, R-EDL, and DAEDL, which require Fisher information regularisation, extra hyperparameters, or post-hoc Gaussian fitting. It also matches or surpasses $\mathcal{F}$-EDL (Yoon & Kim, 2025) without requiring spectral normalisation (Miyato et al., 2018) or additional MLP layers.

**CIFAR-10-LT.** Table 2 evaluates long-tail settings, better reflecting real-world scenarios. Without any dataset-specific modification, DAPPr achieves the best accuracy and AUPR

*Table 2.* AUPR (↑) on CIFAR-10-LT (long-tailed) setting. Conf.: confidence estimation using aleatoric uncertainty. OOD detection using epistemic uncertainty against SVHN and CIFAR-100. Baseline results are obtained from (Yoon & Kim, 2025).

| Method | CIFAR-10 ($\rho = 0.01$) | | | | CIFAR-10 ($\rho = 0.1$) | | | |
| | Test Acc. | Conf. | OOD Detection | | Test Acc. | Conf. | OOD Detection | |
| | | | SVHN | CIFAR-100 | | | SVHN | CIFAR-100 |
|---|---|---|---|---|---|---|---|---|
| MC Drop | $39.22_{\pm3.1}$ | $63.62_{\pm2.7}$ | $33.33_{\pm1.7}$ | $54.17_{\pm1.1}$ | $70.87_{\pm3.0}$ | $89.82_{\pm2.3}$ | $37.37_{\pm1.4}$ | $61.18_{\pm1.3}$ |
| EDL | $42.62_{\pm2.7}$ | $82.63_{\pm1.7}$ | $51.99_{\pm3.8}$ | $66.86_{\pm0.9}$ | $79.09_{\pm0.4}$ | $95.36_{\pm0.1}$ | $72.18_{\pm2.1}$ | $80.09_{\pm0.7}$ |
| $\mathcal{I}$-EDL | $57.88_{\pm1.3}$ | $84.10_{\pm1.3}$ | $52.85_{\pm6.8}$ | $69.19_{\pm1.3}$ | $84.86_{\pm0.1}$ | $97.31_{\pm0.2}$ | $79.83_{\pm3.9}$ | $83.50_{\pm0.4}$ |
| R-EDL | $63.36_{\pm1.0}$ | $78.34_{\pm1.0}$ | $48.71_{\pm7.1}$ | $64.20_{\pm1.4}$ | $85.35_{\pm0.2}$ | $94.35_{\pm0.2}$ | $60.58_{\pm5.0}$ | $69.53_{\pm1.6}$ |
| DAEDL | $63.36_{\pm1.4}$ | $82.15_{\pm1.0}$ | $51.03_{\pm5.6}$ | $65.31_{\pm1.2}$ | $84.95_{\pm0.4}$ | $95.22_{\pm0.4}$ | $69.40_{\pm4.5}$ | $74.56_{\pm1.7}$ |
| $\mathcal{F}$-EDL | $63.73_{\pm1.4}$ | $85.99_{\pm1.7}$ | $62.56_{\pm2.8}$ | $70.18_{\pm2.0}$ | $85.46_{\pm0.2}$ | $97.60_{\pm0.1}$ | $85.36_{\pm1.5}$ | $83.64_{\pm0.7}$ |
| **DAPPr** | $\mathbf{68.81_{\pm0.9}}$ | $\mathbf{87.01_{\pm1.3}}$ | $\mathbf{64.80_{\pm3.6}}$ | $\mathbf{71.77_{\pm1.3}}$ | $\mathbf{86.37_{\pm0.3}}$ | $\mathbf{97.69_{\pm0.1}}$ | $\mathbf{85.63_{\pm2.0}}$ | $\mathbf{84.32_{\pm0.5}}$ |

*Table 3.* AUPR (↑) results on CUB-200-2011 and Stanford Dogs. Conf.: confidence estimation using aleatoric uncertainty. OOD detection using epistemic uncertainty against ImageNet-O, DTD and Places365. Baseline results are reproduced by us.

| Method | CUB-200-2011 | | | | | Stanford Dogs | | | | |
| | Test Acc. | Conf. | OOD Detection | | | Test Acc. | Conf. | OOD Detection | | |
| | | | ImageNet-O | DTD | Places365 | | | ImageNet-O | DTD | Places365 |
|---|---|---|---|---|---|---|---|---|---|---|
| CE | $47.17_{\pm0.6}$ | $80.26_{\pm0.6}$ | $85.21_{\pm0.6}$ | $65.16_{\pm1.3}$ | $36.69_{\pm1.1}$ | $56.35_{\pm0.5}$ | $84.63_{\pm0.6}$ | $94.32_{\pm0.4}$ | $82.29_{\pm1.3}$ | $66.16_{\pm1.9}$ |
| MC Drop | $40.43_{\pm0.7}$ | $73.61_{\pm0.8}$ | $81.40_{\pm0.6}$ | $60.02_{\pm1.3}$ | $31.33_{\pm1.0}$ | $49.84_{\pm1.0}$ | $79.99_{\pm0.8}$ | $90.07_{\pm0.6}$ | $73.26_{\pm1.0}$ | $51.51_{\pm2.0}$ |
| DUQ | $1.55_{\pm0.1}$ | $3.56_{\pm0.1}$ | $69.31_{\pm1.4}$ | $51.25_{\pm1.1}$ | $14.38_{\pm2.0}$ | $12.88_{\pm0.3}$ | $28.26_{\pm0.5}$ | $74.74_{\pm0.2}$ | $43.98_{\pm0.9}$ | $22.97_{\pm1.2}$ |
| KL-PN | $19.43_{\pm1.9}$ | $55.77_{\pm4.7}$ | $74.30_{\pm1.1}$ | $48.27_{\pm1.2}$ | $19.92_{\pm1.4}$ | $31.61_{\pm0.2}$ | $68.04_{\pm0.7}$ | $90.37_{\pm0.4}$ | $71.93_{\pm1.1}$ | $50.56_{\pm1.1}$ |
| RKL-PN | $42.63_{\pm1.8}$ | $78.34_{\pm1.6}$ | $82.95_{\pm1.9}$ | $62.49_{\pm2.4}$ | $35.62_{\pm2.5}$ | $50.24_{\pm0.6}$ | $80.17_{\pm0.4}$ | $95.23_{\pm0.2}$ | $\mathbf{82.92_{\pm1.2}}$ | $69.67_{\pm2.8}$ |
| PostNet | $30.25_{\pm4.0}$ | $62.63_{\pm4.9}$ | $83.00_{\pm1.5}$ | $64.18_{\pm3.3}$ | $24.15_{\pm2.1}$ | $37.83_{\pm1.2}$ | $67.17_{\pm0.5}$ | $86.22_{\pm0.3}$ | $76.45_{\pm0.3}$ | $28.73_{\pm1.2}$ |
| NatPN | $41.49_{\pm0.4}$ | $74.42_{\pm0.6}$ | $88.47_{\pm0.5}$ | $65.91_{\pm3.3}$ | $40.23_{\pm0.8}$ | $50.28_{\pm0.9}$ | $79.71_{\pm0.6}$ | $95.11_{\pm0.1}$ | $81.39_{\pm1.5}$ | $66.89_{\pm1.3}$ |
| RS-NN | $46.21_{\pm0.1}$ | $79.96_{\pm0.5}$ | $85.05_{\pm0.3}$ | $61.56_{\pm0.7}$ | $45.10_{\pm0.8}$ | $57.83_{\pm0.4}$ | $85.05_{\pm0.3}$ | $93.31_{\pm0.2}$ | $81.33_{\pm0.4}$ | $67.92_{\pm0.5}$ |
| EDL | $44.01_{\pm0.7}$ | $82.47_{\pm1.0}$ | $85.02_{\pm1.0}$ | $59.98_{\pm2.3}$ | $46.33_{\pm2.0}$ | $53.06_{\pm1.0}$ | $83.97_{\pm0.2}$ | $93.87_{\pm0.2}$ | $77.14_{\pm0.6}$ | $74.28_{\pm0.3}$ |
| $\mathcal{I}$-EDL | $44.05_{\pm1.2}$ | $82.17_{\pm0.8}$ | $84.99_{\pm1.2}$ | $58.87_{\pm2.7}$ | $46.09_{\pm2.5}$ | $53.22_{\pm0.4}$ | $84.20_{\pm0.3}$ | $93.92_{\pm0.3}$ | $77.42_{\pm1.8}$ | $74.32_{\pm0.4}$ |
| DAEDL | $38.25_{\pm0.9}$ | $77.28_{\pm1.0}$ | $85.87_{\pm0.8}$ | $67.30_{\pm1.8}$ | $45.73_{\pm2.2}$ | $53.34_{\pm0.9}$ | $80.38_{\pm0.5}$ | $94.62_{\pm0.2}$ | $82.10_{\pm1.5}$ | $77.79_{\pm0.8}$ |
| $\mathcal{F}$-EDL | $42.54_{\pm0.7}$ | $77.97_{\pm1.2}$ | $85.26_{\pm0.5}$ | $67.67_{\pm1.3}$ | $37.75_{\pm1.1}$ | $53.00_{\pm0.6}$ | $82.99_{\pm0.2}$ | $95.14_{\pm0.3}$ | $82.59_{\pm1.4}$ | $70.69_{\pm1.6}$ |
| R-EDL | $51.83_{\pm0.3}$ | $86.25_{\pm0.2}$ | $89.37_{\pm0.7}$ | $\mathbf{67.83_{\pm1.9}}$ | $55.74_{\pm2.5}$ | $53.69_{\pm0.8}$ | $83.40_{\pm0.4}$ | $95.08_{\pm0.3}$ | $82.03_{\pm0.7}$ | $78.71_{\pm0.6}$ |
| DAPPr | $\mathbf{55.95_{\pm0.5}}$ | $\mathbf{88.22_{\pm0.4}}$ | $\mathbf{89.53_{\pm0.7}}$ | $67.77_{\pm2.5}$ | $\mathbf{59.39_{\pm2.1}}$ | $\mathbf{61.59_{\pm0.8}}$ | $\mathbf{87.89_{\pm0.3}}$ | $\mathbf{95.47_{\pm0.3}}$ | $82.37_{\pm1.1}$ | $\mathbf{78.76_{\pm1.4}}$ |
| Ensemble | $52.57_{\pm0.2}$ | $85.12_{\pm0.1}$ | $88.98_{\pm0.1}$ | $71.30_{\pm0.3}$ | $47.18_{\pm0.2}$ | $61.11_{\pm0.4}$ | $87.83_{\pm0.1}$ | $96.79_{\pm0.1}$ | $85.75_{\pm0.2}$ | $75.89_{\pm0.4}$ |
| +DAPPr | $\mathbf{61.79_{\pm0.3}}$ | $\mathbf{91.37_{\pm0.1}}$ | $\mathbf{91.02_{\pm0.2}}$ | $\mathbf{72.16_{\pm0.4}}$ | $\mathbf{69.26_{\pm0.3}}$ | $\mathbf{67.08_{\pm0.1}}$ | $\mathbf{90.82_{\pm0.1}}$ | $\mathbf{96.97_{\pm0.1}}$ | $\mathbf{86.44_{\pm0.1}}$ | $\mathbf{85.71_{\pm0.1}}$ |

scores at both $\rho$ values, demonstrating superior robustness.

**Fine-Grained Datasets.** Table 3 evaluates fine-grained datasets with many classes and high resolution. CE and RS-NN use $1 - \max_k p_k$ for aleatoric and entropy for epistemic uncertainty. Implementation details are in §C. Our method achieves best accuracy, confidence estimation, and OOD detection on ImageNet-O and Places365, outperforming all EDL-based methods without requiring additional hyperparameters or architectural modifications. It also surpasses RS-NN (Manchingal et al., 2025), which requires extra computation and pre-trained knowledge. Notably, our single-model method outperforms 10-model ensembles on accuracy, confidence estimation, and Places365 OOD detection. Combining with ensembling yields further gains.

**TinyImageNet.** Table 4 shows that DAPPr outperforms all second-order predictors, including EDL and non-EDL methods, across all metrics, with further gains from ensembling.

**Uncertainty quantification on LLMs.** Following IB-EDL (Li et al., 2025), we finetune Mistral-7B (Jiang et al., 2023) on OBQA (Mihaylov et al., 2018) and report OOD AUROC on ARC-C, ARC-E (Clark et al., 2018), and CSQA (Talmor et al., 2019) (Table 5). DAPPr achieves the best accuracy,

NLL, and OOD detection, outperforming all EDL-based methods, while remaining competitive on ECE.

**Additional Results.** Results in §E show DAPPr achieves competitive performance on MNIST (LeCun, 1998), distribution shift detection, and ImageNet (Deng et al., 2009), and produces well-calibrated reliability diagrams. DAPPr also matches cross-entropy in efficiency (§F).

### 5.3. Ablation studies and analysis

**Ablation of regulariser $\mathcal{R}$.** Table 6 evaluates our spurious evidence regulariser by comparing four variants: (1) DAPPr without $\mathcal{R}$, (2) DAPPr with EDL's KL regularisation, (3) EDL with $\mathcal{R}$, and (4) full DAPPr. Removing or replacing $\mathcal{R}$ degrades all metrics, while applying $\mathcal{R}$ to EDL causes near-zero accuracy, confirming that the regulariser is unique to our possibilistic framework. These results show that the gains stem from suppressing spurious wrong-class evidence rather than uniformly reducing confidence, improving discrimination when similar classes compete.

**Ablation on $\lambda$.** Figure 4 shows effect of regularisation strength $\lambda$ on CIFAR-100 and Stanford Dogs. Without regularisation ($\lambda = 0$), OOD detection degrades significantly.

*Table 4.* AUPR (↑) results on TinyImageNet. Conf.: confidence estimation using aleatoric uncertainty. OOD detection using epistemic uncertainty against Image-O, DTD and Places365.

| Method | Test Acc. | Conf. | OOD Detection | | |
|---|---|---|---|---|---|
| | | | ImageNet-O | DTD | Places365 |
| CE | $58.28_{\pm 1.1}$ | $88.51_{\pm 0.6}$ | $90.39_{\pm 0.4}$ | $78.51_{\pm 1.0}$ | $51.65_{\pm 1.4}$ |
| MC Drop | $51.89_{\pm 1.3}$ | $86.14_{\pm 0.7}$ | $88.10_{\pm 0.5}$ | $73.71_{\pm 1.1}$ | $44.45_{\pm 2.2}$ |
| DUQ | $6.68_{\pm 0.2}$ | $18.13_{\pm 0.4}$ | $80.18_{\pm 0.3}$ | $58.12_{\pm 0.5}$ | $27.09_{\pm 0.8}$ |
| KL-PN | $31.44_{\pm 1.6}$ | $74.73_{\pm 3.8}$ | $88.10_{\pm 0.7}$ | $70.92_{\pm 0.6}$ | $42.29_{\pm 1.9}$ |
| RKL-PN | $55.60_{\pm 0.2}$ | $84.25_{\pm 0.4}$ | $88.97_{\pm 0.9}$ | $74.25_{\pm 1.7}$ | $42.00_{\pm 3.4}$ |
| PostNet | $38.59_{\pm 2.1}$ | $72.80_{\pm 2.7}$ | $86.63_{\pm 0.4}$ | $73.42_{\pm 1.2}$ | $33.14_{\pm 2.2}$ |
| NatPN | $43.78_{\pm 0.6}$ | $75.42_{\pm 1.5}$ | $88.28_{\pm 0.6}$ | $75.36_{\pm 1.1}$ | $41.87_{\pm 1.2}$ |
| RSNN | $55.54_{\pm 0.2}$ | $86.94_{\pm 0.1}$ | $90.16_{\pm 0.1}$ | $77.32_{\pm 0.3}$ | $51.35_{\pm 0.6}$ |
| EDL | $44.40_{\pm 0.6}$ | $88.93_{\pm 0.3}$ | $89.66_{\pm 0.2}$ | $75.01_{\pm 0.3}$ | $51.27_{\pm 0.5}$ |
| $\mathcal{I}$-EDL | $44.41_{\pm 0.4}$ | $89.00_{\pm 0.3}$ | $89.62_{\pm 0.5}$ | $75.36_{\pm 0.9}$ | $51.09_{\pm 0.7}$ |
| DAEDL | $47.44_{\pm 1.9}$ | $80.03_{\pm 3.5}$ | $89.40_{\pm 1.2}$ | $73.74_{\pm 5.5}$ | $48.64_{\pm 3.1}$ |
| $\mathcal{F}$-EDL | $49.90_{\pm 0.5}$ | $71.14_{\pm 2.9}$ | $89.63_{\pm 0.4}$ | $79.06_{\pm 1.1}$ | $45.60_{\pm 1.7}$ |
| R-EDL | $49.76_{\pm 0.4}$ | $86.49_{\pm 0.2}$ | $90.69_{\pm 0.2}$ | $79.12_{\pm 0.3}$ | $53.96_{\pm 0.4}$ |
| DAPPr | $58.19_{\pm 0.7}$ | $91.52_{\pm 0.2}$ | $90.93_{\pm 0.2}$ | $79.51_{\pm 1.1}$ | $57.52_{\pm 0.9}$ |
| Ensemble | $63.76_{\pm 0.2}$ | $91.87_{\pm 0.2}$ | $90.82_{\pm 0.1}$ | $80.87_{\pm 0.1}$ | $57.08_{\pm 0.2}$ |
| +DAPPr | $64.05_{\pm 0.3}$ | $92.61_{\pm 0.0}$ | $91.49_{\pm 0.1}$ | $80.18_{\pm 0.5}$ | $61.80_{\pm 0.1}$ |

*Table 5.* Results on Mistral-7B finetuned on OBQA. OOD AUROC (↑) is evaluated on ARC-C, ARC-E and CSQA.

| Method | Acc. | ECE ↓ | NLL ↓ | OOD Detection | | |
|---|---|---|---|---|---|---|
| | | | | ARC-C | ARC-E | CSQA |
| CE | $88.06_{\pm 0.6}$ | $11.29_{\pm 0.3}$ | $0.85_{\pm 0.0}$ | $60.40_{\pm 1.3}$ | $53.30_{\pm 1.1}$ | $63.70_{\pm 1.1}$ |
| MC Drop | $88.07_{\pm 0.6}$ | $11.29_{\pm 0.4}$ | $0.84_{\pm 0.0}$ | $60.39_{\pm 1.3}$ | $53.30_{\pm 1.1}$ | $63.70_{\pm 1.1}$ |
| Ensemble | $88.51_{\pm 0.1}$ | $8.87_{\pm 0.9}$ | $0.67_{\pm 0.0}$ | $60.67_{\pm 0.8}$ | $54.05_{\pm 0.8}$ | $63.80_{\pm 1.0}$ |
| EDL | $87.23_{\pm 1.4}$ | $6.23_{\pm 0.5}$ | $0.47_{\pm 0.0}$ | $77.34_{\pm 6.9}$ | $74.18_{\pm 6.3}$ | $78.28_{\pm 5.8}$ |
| I-EDL | $88.06_{\pm 0.3}$ | $9.63_{\pm 0.7}$ | $0.45_{\pm 0.0}$ | $82.28_{\pm 1.4}$ | $79.42_{\pm 2.3}$ | $82.07_{\pm 1.4}$ |
| R-EDL | $88.33_{\pm 0.9}$ | $5.43_{\pm 0.6}$ | $0.41_{\pm 0.0}$ | $72.85_{\pm 0.6}$ | $67.56_{\pm 0.7}$ | $71.93_{\pm 2.0}$ |
| IB-EDL | $88.73_{\pm 0.7}$ | $2.27_{\pm 0.6}$ | $0.41_{\pm 0.0}$ | $88.58_{\pm 1.0}$ | $94.29_{\pm 0.4}$ | $83.85_{\pm 2.5}$ |
| DAPPr | $89.47_{\pm 0.6}$ | $3.11_{\pm 0.7}$ | $0.36_{\pm 0.0}$ | $90.42_{\pm 1.2}$ | $95.12_{\pm 0.6}$ | $90.18_{\pm 2.3}$ |

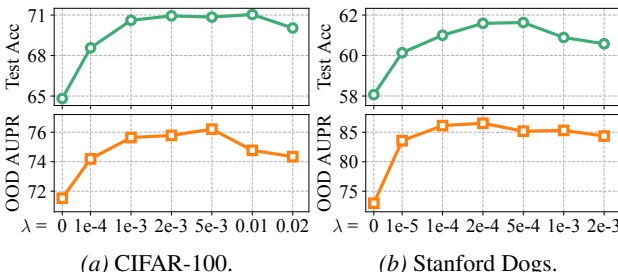

*(a)* CIFAR-100.  *(b)* Stanford Dogs.

*Figure 4.* Test acc and OOD AUPR (↑, averaged over their OOD datasets) for varying $\lambda$ on CIFAR-100 and Stanford Dogs.

Small values such as 2e-4 or 5e-3 achieve strong accuracy and OOD detection. Performance remains stable across this range, indicating low sensitivity to hyperparameter tuning.

**Analysis.** To verify our method properly captures epistemic uncertainty, we vary training data size and measure epistemic uncertainty and accuracy on CIFAR-100. Ideally, epistemic uncertainty should decrease as more data is observed. Figure 5 reveals a flaw in EDL: its epistemic uncertainty increases with more data at certain points, contradicting expected behavior. EDL shows unstable training, with accuracy dropping at 5K datapoints. In contrast, our method consistently decreases epistemic uncertainty as data grows

*Table 6.* Ablation on CUB-200-2011 (CUB) and Stanford Dogs (Dogs). AUPR (↑) for confidence estimation, and OOD detection averaged across three OOD datasets (see §E.5 for more results).

| Dataset | | DAPPr w/o $\mathcal{R}$ | DAPPr w/ KL | EDL w/ $\mathcal{R}$ | DAPPr |
|---|---|---|---|---|---|
| CUB | Acc. | $49.60_{\pm 0.8}$ | $49.84_{\pm 1.2}$ | $0.78_{\pm 0.0}$ | $55.95_{\pm 0.5}$ |
| | Conf. | $83.43_{\pm 0.5}$ | $84.23_{\pm 1.0}$ | $0.90_{\pm 0.1}$ | $88.22_{\pm 0.4}$ |
| | OOD | $53.73_{\pm 1.3}$ | $61.52_{\pm 1.3}$ | $46.57_{\pm 0.8}$ | $72.23_{\pm 1.7}$ |
| Dogs | Acc. | $58.07_{\pm 1.2}$ | $58.92_{\pm 1.4}$ | $1.53_{\pm 0.1}$ | $61.59_{\pm 0.8}$ |
| | Conf. | $86.14_{\pm 0.9}$ | $85.45_{\pm 0.7}$ | $1.84_{\pm 0.1}$ | $87.89_{\pm 0.3}$ |
| | OOD | $72.95_{\pm 1.0}$ | $75.40_{\pm 1.1}$ | $55.79_{\pm 0.6}$ | $85.53_{\pm 0.9}$ |

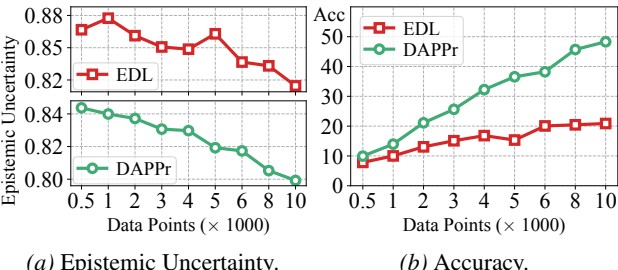

*(a)* Epistemic Uncertainty.  *(b)* Accuracy.

*Figure 5.* Epistemic uncertainty and accuracy on CIFAR-100 with varying training data size. Our method (DAPPr) decreases epistemic uncertainty with more data; EDL does not.

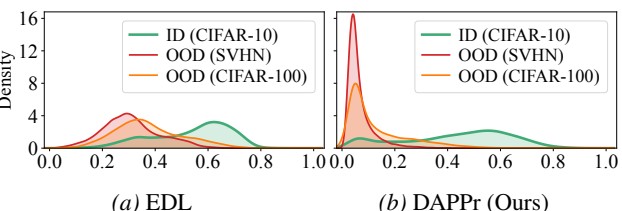

*(a)* EDL  *(b)* DAPPr (Ours)

*Figure 6.* Distribution of normalised $\alpha_0$ on CIFAR-10. DAPPr achieves clear separation; EDL does not.

and maintains stable, higher accuracy throughout. These results confirm that our possibilistic formulation provides a principled foundation for uncertainty modeling, whereas EDL's heuristic interpretation leads to unreliable estimates.

Figure 6 shows the distribution of normalised $\alpha_0$ for ID and OOD samples. EDL produces heavily overlapping distributions, making OOD detection difficult. DAPPr clearly separates ID (high $\alpha_0$) from OOD (low $\alpha_0$), enabling reliable detection of distribution shift.

## 6. Conclusion

We introduced Dirichlet-approximated possibilistic posterior predictions (DAPPr), a principled framework leveraging possibility theory for both theoretically rigorous and computationally efficient epistemic uncertainty quantification. By defining a possibilistic posterior over parameters and projecting it to the prediction space via supremum operators, DAPPr yields a simple training objective with closed-form solutions, achieving competitive or superior performance over SOTA second-order predictors across diverse benchmarks while properly capturing epistemic uncertainty.

## Acknowledgements

This project is supported by the Singapore Ministry of Digital Development and Information under the AI Visiting Professorship Programme (AIVP-2024-004).

## Impact Statement

This paper presents work whose goal is to advance the field of Machine Learning. There are many potential societal consequences of our work, none which we feel must be specifically highlighted here.

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

# A. Proof for Proposition 1

With the log-possibility

$$\log g_{\boldsymbol{\psi}}(\boldsymbol{p} \mid \boldsymbol{x}) = \alpha_0 \log \alpha_0 + \sum_{k=1}^{K} \alpha_k \log \frac{p_k}{\alpha_k},$$

and the cross-entropy loss

$$\ell(\boldsymbol{p}, \boldsymbol{y}) = -\sum_{k=1}^{K} y_k \log p_k,$$

Eq. (13) is derived as follows:

$$
\begin{aligned}
\tilde{\boldsymbol{p}}^* &= \arg\max_{\boldsymbol{p} \in \Delta} \left\{ \log g_{\boldsymbol{\psi}}(\boldsymbol{p} \mid \boldsymbol{x}) + \ell(\boldsymbol{p}, \boldsymbol{y}) \right\} \\
&= \arg\max_{\boldsymbol{p} \in \Delta} \left\{ \alpha_0 \log \alpha_0 + \sum_{k=1}^{K} \alpha_k \log \frac{p_k}{\alpha_k} - \sum_{k=1}^{K} y_k \log p_k \right\} \\
&= \arg\max_{\boldsymbol{p} \in \Delta} \left\{ \sum_{k=1}^{K} (\alpha_k - y_k) \log p_k \right\},
\end{aligned}
\tag{18}
$$

where terms independent of $\boldsymbol{p}$ are omitted.

Since $\boldsymbol{p} \in \Delta$ implies the constraint $\sum_{k=1}^{K} p_k = 1$, we introduce a Lagrange multiplier $\beta$ and form

$$\mathcal{J} = \sum_{k=1}^{K} (\alpha_k - y_k) \log p_k + \beta \left( \sum_{k=1}^{K} p_k - 1 \right). \tag{19}$$

Taking derivatives of Eq. (19) gives the stationarity condition

$$
\begin{aligned}
\frac{\partial \mathcal{J}}{\partial p_k} &= \frac{\alpha_k - y_k}{p_k} + \beta = 0 \\
\Rightarrow \quad p_k &= -\frac{\alpha_k - y_k}{\beta}.
\end{aligned}
\tag{20}
$$

Enforcing $\sum_{k=1}^{K} p_k = 1$ yields

$$
\begin{aligned}
1 = \sum_{k=1}^{K} p_k &= -\frac{1}{\beta} \sum_{k=1}^{K} (\alpha_k - y_k) = -\frac{1}{\beta}(\alpha_0 - 1), \\
\Rightarrow \quad \beta &= -(\alpha_0 - 1).
\end{aligned}
\tag{21}
$$

Substituting Eq. (21) into Eq. (20) gives

$$\tilde{p}_k^* = \frac{\alpha_k - y_k}{\alpha_0 - 1}.$$

Thus, the closed-form expression for the approximate maximizer is

$$\tilde{\boldsymbol{p}}^* = \frac{\boldsymbol{\alpha} - \boldsymbol{y}}{\alpha_0 - 1}. \tag{22}$$

# B. Difference between EDL and DAPPr

Although DAPPr resembles EDL as a lightweight single-pass Dirichlet predictor, the two differ fundamentally in whether epistemic uncertainty is formally defined and targeted in learning.

EDL bypasses the origin of epistemic uncertainty. It assumes a Dirichlet distribution in prediction space, fits labels, and infers uncertainty post-hoc from the concentration parameter $\alpha_0$. Epistemic uncertainty is never its target. It is an interpretation applied after the fact.

DAPPr, by contrast, traces epistemic uncertainty to its source. Since epistemic uncertainty arises from lack of knowledge about parameters, DAPPr defines a possibilistic posterior over parameters, projects it to prediction space via supremum operators, and approximates the projected posterior with a Dirichlet for tractability. DAPPr thus explicitly learns to approximate a well-defined uncertainty target induced by parameter plausibility, a target EDL never establishes.

This theoretical difference has direct empirical consequences. DAPPr's epistemic uncertainty consistently decreases with more data while EDL's pathologically increases (Figure 5). Moreover, our regulariser is not interchangeable with EDL's (Table 6), confirming that the two frameworks are fundamentally different.

## C. Datasets and experiment details

### C.1. Datasets

**MNIST** (LeCun, 1998) consists of 60K training and 10K test examples. We use an 80/20 split for training and validation. KMNIST (Clanuwat et al., 2018) and FashionMNIST (Xiao et al., 2017) serve as OOD datasets. We train with a learning rate of 5e-4, batch size of 64, for 100 epochs.

**CIFAR-10 and CIFAR-10-LT** CIFAR-10 (Krizhevsky et al., 2009) contains 50K training and 10K test images across 10 classes, split 95/5 for training and validation. SVHN (Netzer et al., 2011) and CIFAR-100 serve as OOD datasets. We train VGG16 (Simonyan & Zisserman, 2014) with a learning rate of 5e-4, batch size of 64, for 100 epochs. CIFAR-10-LT is a long-tailed variant with artificially imbalanced class distributions, where the imbalance factor $\rho$, denotes the ratio of head-class to tail-class samples. We evaluate with $\rho \in \{0.01, 0.1\}$.

**CIFAR-100** (Krizhevsky et al., 2009) contains 50K training and 10K test images across 100 classes, split 95/5 for training and validation. SVHN and TinyImageNet (Deng et al., 2009) serve as OOD datasets. We train ResNet-18 (He et al., 2016) with a learning rate of 5e-4, batch size of 64, for 100 epochs.

**CUB-200-2011** (Wah et al., 2011) is a fine-grained bird classification dataset with 200 classes. Following (Jia et al., 2022; Ni et al., 2024), we use 5,394/600/5,794 images for training/validation/testing. ImageNet-O (Hendrycks et al., 2021), DTD (Cimpoi et al., 2014), and Places365 validation set (Zhou et al., 2017) serve as OOD datasets. We train ResNet-50 (He et al., 2016) with a learning rate of 2e-3, weight decay of 1e-4, batch size of 256, for 200 epochs.

**Stanford Dogs** (Khosla et al., 2011) is a fine-grained dog classification dataset with 120 classes. Following (Jia et al., 2022; Ni et al., 2024), we use 10,800/1,200/8,580 images for training/validation/testing. We use the same OOD datasets and training configuration as CUB-200-2011.

**TinyImageNet** (Deng et al., 2009) is a 200-class subset of ImageNet with $64 \times 64$ images, containing 100,000 training images (500 per class) and 10,000 validation images. Following our fine-grained dataset setup, we split the training set 95/5 for training and validation, and use the official validation set for testing. We use the same OOD datasets as CUB-200-2011. We train ResNet-50 (He et al., 2016) with a learning rate of 5e-3, weight decay of 1e-4, batch size of 256, for 100 epochs.

**ImageNet** (Deng et al., 2009) is a large-scale classification dataset with 1,000 classes. As full finetuning is computationally costly, we use a pretrained Masked Autoencoder ViT-B/16 (He et al., 2022) as a frozen feature extractor and append two fully-connected layers (1024 and 512 units with ReLU activations). For OOD detection, we use the same OOD datasets as CUB-200-2011. We train with a learning rate of 1e-3, weight decay of 1e-4, batch size 1024, for 100 epochs.

**OpenBookQA (OBQA)** (Mihaylov et al., 2018) is a question-answering dataset containing 4,957 training, 500 validation, and 500 test questions. We finetune Mistral-7B (Jiang et al., 2023) on the training set and use the validation set for validation. For OOD detection, we use ARC-Challenge (ARC-C) and ARC-Easy (ARC-E) (Clark et al., 2018), and CommonsenseQA (CSQA) (Talmor et al., 2019) as OOD datasets. We follow the same finetuning configuration as IB-EDL (Li et al., 2025) for fair comparison and report mean±std over 5 random seeds.

### C.2. Hyperparameters

Table 7 presents the regularisation hyperparameter $\lambda$ across different datasets. Notably, $\lambda$ remains consistently small across all datasets, requiring minimal tuning, which demonstrates the stability and ease of use of our method. We use two schedule variants: *warmup*, where $\lambda_t = \lambda \cdot \min(1, t/10)$ increases linearly over the first 10 epochs then remains fixed, and *linear*, where $\lambda_t = \lambda \cdot t/T$ increases gradually throughout training, with $t$ the current epoch and $T$ the total number of epochs.

*Table 7.* Hyperparameters used for our regularizer.

| Dataset | MNIST | CIFAR-10 | CIFAR-10-LT | CIFAR-100 | CUB-200-2011 | Stanford Dogs | TinyImageNet | ImageNet | OBQA |
|---|---|---|---|---|---|---|---|---|---|
| $\lambda$ | 1e-5 | 2e-3 | 2e-3 | 5e-3 | 2e-4 | 2e-4 | 5e-3 | 5e-3 | 2e-4 |
| $\lambda$ schedule | warmup | warmup | warmup | warmup | warmup | warmup | linear | linear | warmup |

### C.3. Baseline implementations for fine-grained datasets

**CE**: We train the network with cross-entropy loss. For confidence estimation, we use $1 - \max_k p_k$ as aleatoric uncertainty. For OOD detection, we use the entropy over class probabilities as epistemic uncertainty.

**MC Drop**: After training with cross-entropy, we perform $M = 10$ stochastic forward passes with dropout rate 0.5, obtaining predictions $\{\boldsymbol{p}^{(m)}\}_{m=1}^M$. The averaged prediction is $\bar{\boldsymbol{p}} = \frac{1}{M} \sum_{m=1}^M \boldsymbol{p}^{(m)}$. We use $1 - \max_k \bar{p}_k$ as aleatoric uncertainty for confidence estimation, and $EU = H(\bar{\boldsymbol{p}}) - \frac{1}{M} \sum_{m=1}^M H(\boldsymbol{p}^{(m)})$ as epistemic uncertainty for OOD detection.

**Ensemble**: We train $M = 10$ models from scratch with different random seeds. Predictions are averaged as $\bar{\boldsymbol{p}} = \frac{1}{M} \sum_{m=1}^M \boldsymbol{p}^{(m)}$. Aleatoric and epistemic uncertainties are computed identically to MC Dropout.

**Ensemble+DAPPr**: We train $M$ models using DAPPr loss with different random seeds. The averaged outputs serve as Dirichlet parameters for computing aleatoric and epistemic uncertainties.

**RS-NN**: Following (Manchingal et al., 2025), we use a pretrained ResNet-50 for feature extraction and construct random sets as new classes. We search hyperparameters over {1e-6, 5e-6, 2e-5, 1e-4, 5e-4, 1e-3, 5e-3} and report the best results.

**EDL, $\mathcal{I}$-EDL, DAEDL, $\mathcal{F}$-EDL, R-EDL**: For each EDL-based method, we tune the KL regularisation coefficient along with method-specific hyperparameters (e.g., Fisher information regularizer for $\mathcal{I}$-EDL, evidence offset for R-EDL, and MLP architecture for $\mathcal{F}$-EDL). We report the best results from the hyperparameter sweep.

**DUQ, KL-PN, RKL-PN, PostNet, NatPN**: For each method, we tune method-specific hyperparameters following their original implementations, including the gradient penalty for DUQ, the KL regularisation coefficient for KL-PN and RKL-PN, and the normalising flow architecture for PostNet and NatPN. We report the best results from the hyperparameter sweep.

## D. Experiments for validating the approximation

To verify the approximation in Eq. (11), we conduct a leave-one-out analysis on CIFAR-100 using a pre-trained ResNet-18 model $\boldsymbol{\theta}_0$. For each sample $(\boldsymbol{x}, \boldsymbol{y})$ in the training set, we perform two fine-tuning experiments:

1. **True label fine-tuning:** Fine-tune $\boldsymbol{\theta}_0$ for three epochs on $\mathcal{D} \setminus \{(\boldsymbol{x}, \boldsymbol{y})\}$ while forcing the model to fit $(\boldsymbol{x}, \boldsymbol{y})$ by including it in every batch during training, yielding model $\boldsymbol{\theta}_{\text{true}}$ and loss $L_{\text{true}} = L(\boldsymbol{\theta}_{\text{true}}; \mathcal{D} \setminus \{(\boldsymbol{x}, \boldsymbol{y})\})$.

2. **Perturbed label fine-tuning:** Fine-tune $\boldsymbol{\theta}_0$ identically, but replace the label of $\boldsymbol{x}$ with a randomly sampled soft label $\boldsymbol{p} \in \Delta^{K-1}$ for $\boldsymbol{x}$, yielding model $\boldsymbol{\theta}_{\boldsymbol{p}}$ and loss $L_{\boldsymbol{p}} = L(\boldsymbol{\theta}_{\boldsymbol{p}}; \mathcal{D} \setminus \{(\boldsymbol{x}, \boldsymbol{y})\})$.

For each sample, we compute the maximum loss deviation $S_{\boldsymbol{x}} = \max_{\boldsymbol{p}} |L_{\boldsymbol{p}} - L_{\text{true}}|$ over multiple random perturbations. Figure 7 plots $(L_{\text{true}}, S_{\boldsymbol{x}})$ for all samples. The results show that $S_{\boldsymbol{x}}$ is negligibly small relative to $L_{\text{true}}$ (e.g., $S_{\boldsymbol{x}} \approx 0.25$ versus $L_{\text{true}} \approx 320$, validating that perturbing a single sample's label has minimal impact on the overall loss, a key assumption underlying our approximation.

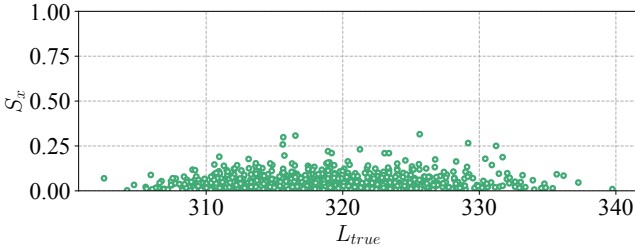

*Figure 7.* $x$ axis: $L_{\text{true}}$. $y$-axis: $S_x$

# E. Additional results

## E.1. Results on MNIST

We use ConvNet (3 convolutional + 3 dense layers) for MNIST, with KMNIST (Clanuwat et al., 2018) and FashionMNIST (Xiao et al., 2017) as OOD datasets (Table 8). Despite using only a simple loss with a regulariser, DAPPr achieves the best AUPR for confidence estimation and OOD detection while remaining competitive in accuracy, without the multiple hyperparameters or computational overhead required by DUQ, PostNet, and EDL-based methods.

*Table 8.* AUPR (↑) results on MNIST. Conf.: confidence estimation using aleatoric uncertainty. OOD detection uses epistemic uncertainty against KMNIST and FMNIST. Baseline results from (Chen et al., 2024) where available; others are our reproduction.

| Method | Test Acc. | Conf. | OOD Detection | |
| | | | KMNIST | FMNIST |
|---|---|---|---|---|
| MC Dropout | $99.26_{\pm 0.0}$ | $99.98_{\pm 0.0}$ | $94.00_{\pm 0.1}$ | $96.56_{\pm 0.3}$ |
| DUQ | $98.65_{\pm 0.1}$ | $99.97_{\pm 0.0}$ | $98.52_{\pm 0.1}$ | $97.92_{\pm 0.6}$ |
| KL-PN | $99.01_{\pm 0.0}$ | $99.92_{\pm 0.0}$ | $93.39_{\pm 1.0}$ | $98.16_{\pm 0.0}$ |
| RKL-PN | $99.21_{\pm 0.0}$ | $99.67_{\pm 0.0}$ | $53.76_{\pm 3.4}$ | $72.18_{\pm 3.6}$ |
| PostNet | $\mathbf{99.34_{\pm 0.0}}$ | $99.98_{\pm 0.0}$ | $94.59_{\pm 0.3}$ | $97.24_{\pm 0.3}$ |
| EDL | $98.22_{\pm 0.3}$ | $\mathbf{99.99_{\pm 0.0}}$ | $96.31_{\pm 2.0}$ | $98.08_{\pm 0.4}$ |
| $\mathcal{I}$-EDL | $99.21_{\pm 0.0}$ | $99.98_{\pm 0.0}$ | $98.33_{\pm 0.2}$ | $98.86_{\pm 0.2}$ |
| R-EDL | $99.33_{\pm 0.0}$ | $\mathbf{99.99_{\pm 0.0}}$ | $98.69_{\pm 0.2}$ | $99.29_{\pm 0.1}$ |
| $\mathcal{F}$-EDL | $99.30_{\pm 0.2}$ | $99.93_{\pm 0.0}$ | $98.74_{\pm 0.3}$ | $99.31_{\pm 0.2}$ |
| DAPPr | $99.26_{\pm 0.1}$ | $\mathbf{99.99_{\pm 0.0}}$ | $\mathbf{98.81_{\pm 0.2}}$ | $\mathbf{99.55_{\pm 0.1}}$ |

## E.2. Results on distribution shift detection

Table 9 shows that distribution shift detection improves as corruption severity increases (i.e., as data diverges further from the training distribution). CE: cross-entropy using $1 - \max_k p_k$ for aleatoric uncertainty. Our method achieves the best AUPR at severity levels 2–5.

*Table 9.* AUPR (↑) for detecting distribution shift from CIFAR-10 to CIFAR-10-C (Hendrycks & Gimpel, 2017) using aleatoric uncertainty. $\mathcal{C} \in \{1, 2, 3, 4, 5\}$ denotes corruption severity levels. Results are averaged over 19 corruption types. Baseline results are from (Yoon & Kim, 2025).

| Method | $\mathcal{C} = 1$ | $\mathcal{C} = 2$ | $\mathcal{C} = 3$ | $\mathcal{C} = 4$ | $\mathcal{C} = 5$ |
|---|---|---|---|---|---|
| CE | $56.39_{\pm 0.7}$ | $61.88_{\pm 1.1}$ | $65.86_{\pm 1.3}$ | $69.91_{\pm 1.5}$ | $75.01_{\pm 1.8}$ |
| EDL | $54.76_{\pm 0.3}$ | $59.01_{\pm 0.4}$ | $62.46_{\pm 0.5}$ | $65.87_{\pm 0.6}$ | $70.21_{\pm 0.8}$ |
| $\mathcal{I}$-EDL | $56.33_{\pm 0.2}$ | $61.52_{\pm 0.5}$ | $65.44_{\pm 0.5}$ | $69.45_{\pm 0.5}$ | $74.56_{\pm 0.5}$ |
| R-EDL | $57.37_{\pm 0.5}$ | $62.20_{\pm 1.0}$ | $65.74_{\pm 1.4}$ | $69.33_{\pm 1.9}$ | $73.58_{\pm 2.6}$ |
| DAEDL | $57.89_{\pm 0.3}$ | $63.23_{\pm 0.4}$ | $67.53_{\pm 0.4}$ | $72.21_{\pm 0.4}$ | $77.74_{\pm 0.4}$ |
| $\mathcal{F}$-EDL | $\mathbf{59.01_{\pm 0.8}}$ | $65.11_{\pm 0.7}$ | $69.48_{\pm 0.5}$ | $73.88_{\pm 0.3}$ | $78.72_{\pm 0.4}$ |
| DAPPr | $58.81_{\pm 0.2}$ | $\mathbf{65.20_{\pm 0.3}}$ | $\mathbf{69.83_{\pm 0.5}}$ | $\mathbf{74.29_{\pm 0.3}}$ | $\mathbf{79.73_{\pm 0.6}}$ |

## E.3. Results on ImageNet

As full finetuning on ImageNet is computationally costly, we use a pretrained Masked Autoencoder ViT-B/16 (He et al., 2022) as a frozen feature extractor with two appended fully-connected layers (1024 and 512 units, ReLU). We compare against strong baselines including RPN and NatPN; $\mathcal{F}$-EDL and DAEDL are excluded due to NaN errors. As shown in Table 10, DAPPr achieves the best performance across all metrics, demonstrating that our method scales effectively to 1000-class classification.

## E.4. Reliability diagrams

Figure 8 shows reliability diagrams on CIFAR-10 and Stanford Dogs comparing DAPPr against EDL-based methods (EDL, $\mathcal{I}$, $\mathcal{F}$-EDL, R-EDL) and non-EDL second-order predictors (DUQ, KL-PN, RKL-PN, PostNet, NatPN, RSNN), where a well-calibrated model follows the ideal diagonal $y = x$. On CIFAR-10, DAPPr tracks the diagonal closely, particularly in the high-confidence region (0.7-1.0), where reliable predictions matter most. On Stanford Dogs, most competing methods are generally overconfident, with accuracy falling consistently below their confidence, while DAPPr remains well-calibrated

*Table 10.* AUPR ($\uparrow$) results on ImageNet. Conf.: confidence estimation via aleatoric uncertainty. OOD detection via epistemic uncertainty against ImageNet-O, DTD, and Places365, averaged over the three datasets.

| Method | Test Acc. | Conf. | OOD |
|---|---|---|---|
| RPN | $53.47_{\pm0.5}$ | $82.62_{\pm0.3}$ | $87.13_{\pm0.2}$ |
| NatPN | $60.18_{\pm0.4}$ | $90.56_{\pm0.2}$ | $89.34_{\pm0.3}$ |
| EDL | $49.93_{\pm0.3}$ | $90.34_{\pm0.1}$ | $89.15_{\pm0.1}$ |
| $\mathcal{I}$-EDL | $50.20_{\pm0.4}$ | $90.49_{\pm0.1}$ | $89.29_{\pm0.1}$ |
| R-EDL | $61.15_{\pm0.8}$ | $90.97_{\pm0.3}$ | $90.08_{\pm0.2}$ |
| DAPPr | $\mathbf{63.43_{\pm0.1}}$ | $\mathbf{91.43_{\pm0.2}}$ | $\mathbf{90.71_{\pm0.1}}$ |

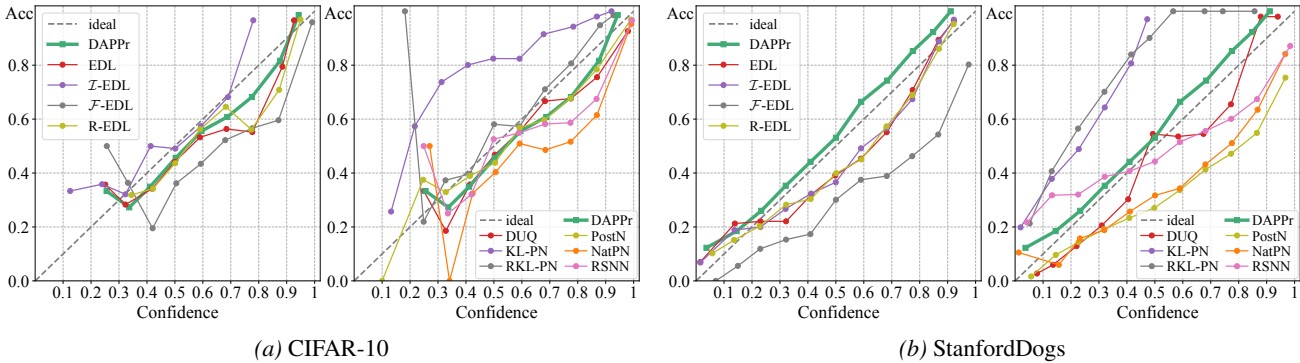

*(a)* CIFAR-10        *(b)* StanfordDogs

*Figure 8.* Reliability diagrams on CIFAR-10 and Stanford Dogs comparing DAPPr against EDL-based methods (left) and non-EDL second-order predictors (right). A well-calibrated model follows the diagonal $y = x$ (grey dashed).

and closely follows the diagonal. These results confirm that DAPPr produces reliable confidence estimates across both standard and fine-grained settings.

### E.5. Ablation study on the regulariser for CIFAR-10-LT and CIFAR-100

We additionally conduct an ablation study on our regulariser $\mathcal{R}$ on CIFAR-10-LT ($\rho = 0.01$) and CIFAR-100, Table 11 shows that shows that removing or replacing our regulariser with EDL's KL regularisation degrades performance, confirming that the regulariser is specific to our possibilistic formulation and not a generic confidence penalty.

*Table 11.* Ablation on CIFAR-10-LT ($\rho = 0.01$) and CIFAR-100. AUPR ($\uparrow$) for confidence estimation and OOD detection averaged across their corresponding OOD datasets.

| Method | CIFAR-10-LT ($\rho = 0.01$) | | | CIFAR-100 | | |
|---|---|---|---|---|---|---|
| | Test Acc. | Conf. | OOD | Test Acc. | Conf. | OOD |
| DAPPr w/o $\mathcal{R}$ | $66.74_{\pm0.1}$ | $83.35_{\pm1.1}$ | $57.49_{\pm1.7}$ | $64.45_{\pm0.3}$ | $90.52_{\pm0.2}$ | $71.52_{\pm1.2}$ |
| DAPPr w/ KL | $67.58_{\pm0.8}$ | $85.59_{\pm0.6}$ | $65.82_{\pm2.7}$ | $69.54_{\pm0.5}$ | $93.66_{\pm1.1}$ | $73.73_{\pm1.2}$ |
| EDL w/ $\mathcal{R}$ | $56.10_{\pm0.8}$ | $80.30_{\pm1.8}$ | $58.84_{\pm2.7}$ | $42.32_{\pm2.4}$ | $90.02_{\pm1.6}$ | $60.71_{\pm2.5}$ |
| DAPPr | $68.81_{\pm0.9}$ | $87.01_{\pm1.3}$ | $68.29_{\pm2.5}$ | $70.85_{\pm0.2}$ | $94.39_{\pm0.1}$ | $76.22_{\pm1.6}$ |

## F. Cost and latency

Table 12 reports cost and latency on Stanford Dogs. DUQ, KL-PN, and RKL-PN require substantially more memory and training time, while PostNet and DAEDL incur significant inference overhead. DAPPr matches standard cross-entropy training in all efficiency metrics, confirming that principled uncertainty quantification need not come at additional computational cost.

*Table 12.* Cost and latency on Stanford Dogs, averaged over 5 runs on a V100 32G GPU. All methods use a batch size of 128 for fair comparison, as DUQ, KL-PN, and RKL-PN run out of memory with a batch size of 256.

| Method | #Para.(M) | GPU Mem (G) | Train 1 Epoch (sec) | Inference (sec) |
|---|---|---|---|---|
| CE | 22.7 | 12.3 | 36 | 52 |
| DUQ | 30.2 | 27.3 | 248 | 52 |
| KL-PN | 22.7 | 26.4 | 86 | 52 |
| RKL-PN | 22.7 | 26.4 | 86 | 52 |
| PostNet | 22.5 | 12.4 | 106 | 96 |
| NatPN | 22.5 | 12.3 | 37 | 52 |
| RSNN | 22.9 | 12.3 | 37 | 56 |
| EDL | 22.7 | 12.3 | 37 | 52 |
| $\mathcal{I}$-EDL | 22.7 | 12.3 | 37 | 52 |
| DAEDL | 22.7 | 12.3 | 37 | 264 |
| $\mathcal{F}$-EDL | 24.8 | 12.4 | 37 | 61 |
| R-EDL | 22.7 | 12.3 | 37 | 52 |
| DAPPr | 22.7 | 12.3 | 37 | 52 |

