# OpenReview forum: "Possibilistic Predictive Uncertainty for Deep Learning"
_ICML.cc/2026/Conference — ICML 2026 regular_

### Official Review · Reviewer_Ro7a · 2026-02-18

**Soundness:** 3
**Presentation:** 3
**Significance:** 3
**Originality:** 3
**Overall Recommendation:** 4
**Confidence:** 3

**Summary:**

This paper introduces Dirichlet-approximated possibilistic posterior predictions (DAPPr). DAPPr applies the possibility theory framework for neural-network model parameters. Under the Bayesian perspective, the possibilistic posterior over the parameter provides epistemic uncertainty with a closed-form solution. Theoretically, DAPPr is derived by a tractable implementation with closed-form solutions. Empirically, DAPPr achieves competitive or superior uncertainty quantification (UQ) while maintaining computational efficiency with other baselines.

**Compliance With Llm Reviewing Policy:**

Affirmed.

**Final Justification:**

The rebuttal addressed my main concerns about calibration error.

**Key Questions For Authors:**

1. Is DAPPr limited by the number of classes? Have the authors considered deploying on a dataset with a larger number of classes (e.g., ImageNet)?
2. How does DAPPr latency compare to other Second-order predictive models (e.g., EDL, PostNet, Natural Posterior Networks, DUQ, etc.)?

**Limitations:**

The limitations (e.g., regularization hyperparameter, additional training cost, etc.) are not adequately discussed.

**Strengths And Weaknesses:**

**Strengths**:
- The proposed method is developed rigorously and is novel in applying the possibility theory framework for a neural network.
- This paper is generally well-written and is clear to understand the important aspects.
- Empirical results are quite consistent across a diverse range of baselines.

**Weaknesses**:
- The evaluation is missing key metrics, including computational efficiency and expected calibration error.
- The proposed method only considers the Dirichlet possibility function and is only designed for classification tasks.
- The baselines are not consistent across different datasets, e.g., PostNet and Ensembles are missing in the CIFAR dataset.
- The baselines are also missing other second-order predictive methods that are discussed in related works.

---

> ### Author Rebuttal · Authors · 2026-03-30
>
> **Thank you for reviews.**
>
> All results (W1,W3,W4,Q1) are over 5 runs (std omitted for brevity-we can provide std in discussion)
> # W1: ECE/efficiency.
> ECE ↓ on CIFAR-10/StanfordDogs:
> Method|CIFAR-10|StanfordDogs|Acc.↑
> :-|:-:|:-:|:-:
> CE|6.9|28.8|74
> MC Drop|5.9|17.1|66
> DUQ|2.4|10.2|50
> KL-PN|18.2|21.8|27
> RKL-PN|4.9|29.3|67
> PostNet|**2.4**|17.3|63
> NatPN|5.1|16.1|70
> RSNN|4.7|11.7|74
> EDL|2.6|7.4|68
> I-EDL|16.7|7.7|71
> DAEDL|3.6|7.5|72
> F-EDL|4.8|21.4|72
> R-EDL|4.8|7.9|71
> DAPPr|4.6|7.5|76
> Ensemble|3.1|10.9|77
> Ensemble+DAPPr|4.2|**7.3**|**80**
>
> **DAPPr has good ECE. As ECE is not stable metric** due to Bin Sensitivity, Cancellation (errors can cancel out in a bin due to abs. diff.), Majority Class Trap (low ECE yet poor Acc.), **it must be viewed jointly with Accuracy** (we average Acc. over 2 datasets for brevity).
> \
> \
> Table (StanfordDogs) below (BS=128; DUQ/KL-PN/RKL-PN exceed memory at BS=256, avg. of 5 runs, V100 32G). Inference time covers confidence estimation/OOD detection on 3 datasets.
> Method|Para.(M)|GPU Mem (G)|Train/Epoch (sec)|Infer. (sec)
> :-|:-:|:-:|:-:|:-:
> CE|22.7|12.3|36|52
> DUQ|30.2|27.3|248|52
> KL-PN|22.7|26.4|86|52
> RKL-PN|22.7|26.4|86|52
> PostNet|22.5|12.4|106|96
> NatPN|22.5|12.3|37|52
> RSNN|22.9|12.3|37|56
> EDL|22.7|12.3|37|52
> I-EDL|22.7|12.3|37|52
> DAEDL|22.7|12.3|37|264
> F-EDL|24.8|12.4|37|61
> R-EDL|22.7|12.3|37|52
> **DAPPr**|22.7|12.3|37|52
>
> - DUQ/KL-PN/RKL-PN use a lot more Mem/train time
> - PostNet/DAEDL have large inference overhead
> - DAPPr matches CE in efficiency & improves UQ
> # W2: Scope.
> Our framework is not limited to classification:
> - Possibilistic posterior, supremum projection, and $⁡D_\max$ matching apply to general possibility functions on parameter space.
> - We employ Dirichlet for classification due to tractability, yielding a closed-form surrogate under cross-entropy on the simplex.
> - **Other settings need non-trivial design of suitable possibility families with tractable surrogate**: we leave possibilistic Gaussian+regression for future work. DAPPr is a foundation for extensions. We will clarify it.
> # W3/W4: Other baselines.
> We reproduce baselines with tuned hyperparam. DAPPr outperforms non-EDL methods.
>
> AUPR ↑ CIFAR-10 (OOD: SVHN, CIFAR-100):
> Method|Acc|Conf.|SVHN|CIFAR-100
> :-|:-:|:-:|:-:|:-:
> DUQ|88.79|98.35|86.62|85.28
> KL-PN|23.94|35.22|49.56|58.18
> RKL-PN|85.48|97.12|65.36|69.37
> PostNet|88.54|98.38|82.34|82.25
> NatPN| 90.29|98.58|87.12|85.51
> RSNN|91.91|99.19|89.68|88.94
> DAPPr|92.00|99.23|91.72|89.39
> Ensemble|**93.82**|99.49|86.90|89.92|91.47|90.78
> Ensemble+DAPPr|93.63|**99.51**|**94.73**|**91.45**|**92.67**|**91.07**
>
> AUPR CIFAR-100 (SVHN, TinyImageNet):
> Method|Acc|Conf.|SVHN|TinyImageNet
> :-|:-:|:-:|:-:|:-:
> DUQ|57.93|84.30|64.90|71.64
> KL-PN|41.72|82.42|47.03|67.33
> RKL-PN|69.92|93.12|67.92|74.60
> PostNet|54.69|85.56|54.92|61.68
> NatPN|65.95|90.84|68.98|74.94
> RSNN|68.60|92.73|73.51|78.08
> DAPPr|70.85|94.39|73.32|79.11
> Ensemble|**75.48**|94.89|**75.09**|80.02|
> Ensemble+DAPPr|74.38|**95.38**|74.31|**80.88**|
>
> AUPR CUB-200-2011 (ImageNet-O, DTD, Places365):
> Method|Acc|Conf.|ImageNet-O|DTD|Places365
> :-|:-:|:-:|:-:|:-:|:-:
> DUQ|1.55|3.56|69.31|51.25|14.38
> KL-PN|19.43|55.77|74.30|48.27|19.92
> RKL-PN|42.63|78.34|82.95|62.49|35.62
> PostNet|30.25|62.63|83.00|64.18|24.15
> NatPN|41.49|74.42|88.47|65.91|40.23
> DAPPr|**55.95**|**88.22**|**89.53**|**67.77**|**59.39**
>
> AUPR StanfordDogs:
> Method|Acc|Conf.|ImageNet-O|DTD|Places365
> :-|:-:|:-:|:-:|:-:|:-:
> DUQ|12.88|28.26|74.74|43.98|22.97
> KL-PN|31.61|68.04|90.37|71.93|50.56
> RKL-PN|50.24|80.17|95.23|**82.92**|69.67
> PostNet|37.83|67.17|86.22|76.45|28.73
> NatPN|50.28|79.71|95.11|81.39|66.89
> DAPPr|**61.59**|**87.89**|**95.47**|82.37|**78.76**
> # Q1: Use more classes.
> **TinyImageNet (200 classes, ResNet-50 trained from scratch):** DAPPr outperforms all baselines.
> Method|Acc|Conf.|ImageNet-O|DTD|Places365
> :-|:-:|:-:|:-:|:-:|:-:
> CE|58.28|88.51|90.39|78.51|51.65
> MC Drop|51.89|86.14|88.10|73.71|44.45
> DUQ|6.68|18.13|80.18|58.12|27.09
> KL-PN|31.44|74.73|88.10|70.92|42.29
> RKL-PN|55.60|84.25|88.97|74.25|42.00
> PostNet|38.59|72.80|86.63|73.42|33.14
> NatPN|43.78|75.42|88.28|75.36|41.87
> RSNN|55.54|86.94|90.16|77.32|51.35
> EDL|44.40|88.93|89.66|75.01|51.27
> I-EDL|44.41|89.00|89.62|75.36|51.09
> DAEDL|47.44|80.03|89.40|73.74|48.64
> F-EDL|49.90|71.14|89.63|79.06|45.60
> R-EDL|49.76|86.49|90.69|79.12|53.96
> DAPPr|58.19|91.52|90.93|79.51|57.52
> Ensemble|63.76|91.87|90.82|**80.87**|57.08
> Ensemble+DAPPr|**64.05**|**92.61**|**91.49**|80.18|**61.80**
>
> **ImageNet (1000 classes):** As full ImageNet training is costly, we use pretrained Masked Autoencoder ViT-B/16 as feature extractor, add 2 extra FC layers (1024, 512, ReLU, BS: 1024, 100 epochs). Strong RPN/NatPN are selected. F-EDL/DAEDL are excluded due to NaN errors. DAPPr is best/scales to 1000 classes.
> Method|Acc|Conf.|OOD
> :-|:-:|:-:|:-:
> RPN|53.47|82.62|87.13
> NatPN|60.18|90.56|89.34
> EDL|49.93|90.34|89.15
> I-EDL|50.20|90.49|89.29
> R-EDL|61.15|90.97|90.08
> DAPPr|**63.43**|**91.43**|**90.71**
> # L1: Hyperparam.
> Tab. 7/Fig. 1 show λ is small/stable on datasets

---

> > ### Author Rebuttal · Reviewer_Ro7a · 2026-03-31
> >
> > I thank the authors for the rebuttal. While I appreciated your additional results, the clarification and explanation are still not fully resolved for me.
> >
> > **W1.** I agree that ECE may be sensitive to the number of bins; yet, it is a standard calibration evaluation metric for neural networks. There are also many variants of measuring calibration without bins, e.g., Kolmogorov-Smirnov (KS) error [1], ECE with KDE [2], etc.
> >
> > From the first table, although your accuracy is higher, the ECE is worse than others in CIFAR-10. By the definition of calibration error, this result implies that your confidence quality is worse than that of others. I would suggest that the authors show the reliability diagram. I think this helps to understand more about the difference between accuracy and confidence quality.
> >
> > [1] Gupta et al., Calibration of neural networks using splines, ICLR, 2021.
> >
> > [2] Popordanoska et al., A Consistent and Differentiable Lp Canonical Calibration Error Estimator, NeurIPS, 2022.
> >
> > **W2.** Thanks for your clarification. The explanation seems that the proposed method is currently non-trivially extendable to regression. If it is extendable, I think including a small extension to other settings would improve the experiments.
> >
> > **W3-4, Q1.** Thanks for providing many experimental results. It seems the results are not really consistent. I think adding these results and more analysis about how, when, and why the method is better or worse than others would strengthen the paper in the next version.
> >
> > ---
> >
> > **Follow-up for author's response below**: I sincerely thank the authors for your reply and for providing several new experiments. These results and the explanation are convincing for me to increase my original rating. Hence, I increased my score from 3 to 4.
> >
> > Best,
> >
> > Reviewer.

---

> > > ### Author Response · Authors · 2026-04-03
> > >
> > > We thank Rev. for deep insights/fast feedback.
> > >
> > > # W1: Provide ECE(KDE) & KS:
> > > DAPPr achieves competitive scores.
> > > ||CIFAR-10||StanfordDogs||
> > > -|-|-|-|-
> > > **Method**|ECE(KDE)|KS|ECE(KDE)|KS
> > > CE|0.171|5.9|1.112|28.5
> > > MC Drop|0.152|5.9|1.037|21.0
> > > DUQ|0.296|3.1|1.877|12.0
> > > KL-PN|0.778|16.5|0.968|21.9
> > > RKL-PN|0.421|5.8|1.625|29.4
> > > PostN|0.221|3.0|1.360|18.9
> > > NatPN|0.154|5.5|0.931|17.8
> > > RSNN|0.145|3.8|0.952|10.3|
> > > EDL|0.256|2.5|0.952|5.7
> > > I-EDL|0.429|11.3|0.955|5.7
> > > DAEDL|0.144|4.9|0.930|5.4
> > > F-EDL|**0.143**|4.8|0.943|21.8|
> > > R-EDL|0.183|2.3|1.068|6.0
> > > **DAPPr (our)**|**0.143**|**1.9**|**0.925**|**5.2**
> > > Ensemble|0.119|2.3|0.927|11.8
> > > **Ensemble+DAPPr (our)**|**0.103**|**1.8**|**0.812**|**5.1**
> > > # W1b: Reliability diagram
> > > - Kindly see https://is.gd/vS6HEz
> > >
> > > - Anonymous links to rebuttal figs are allowed (ICML'26 policy). **If not allowed in follow-ups, please do not open.**
> > >
> > > - The best reliability is when the curve is close to the linear $y=x$ function (accuracy matches confidence).
> > >
> > > - CIFAR-10: **DAPPr tracks the diagonal very close esp. in 0.7–1.0 conf.**
> > >
> > > - StanfordDogs: competitors tend to be overconfident. **DAPPr has conservative underconfidence/matches the diagonal well.**
> > >
> > > # W2: Extend to Regression
> > > Let regression version of DAPPr use isotropic Gaussian & $L^2$ loss. Set the per-sample loss as $\ell(z,y)=\frac{1}{2}(z-y)^2$
> > > $$
> > > g_\psi(z|x) = \exp(-\tfrac{\gamma_\psi(x)}{2} (z-\mu_\psi(x))^2),
> > > $$
> > > where $\mu_\psi(x)\in\mathbb{R}$ and $\gamma_\psi(x)>0$. Assume $\gamma_\psi(x)>\beta$. Minimizing $D_\max(g_\psi(\cdot|x)\|\|g_x^*(\cdot|D) )$ amounts to minimizing
> > > $$
> > > \mathcal{J}\_\psi(x,y)=\frac{\beta\gamma_\psi(x)}{2(\gamma_\psi(x)-\beta)} \|\mu_\psi(x)-y\|^2,
> > > $$
> > > which is well defined when
> > > $$
> > > \gamma_\psi(x)=\beta+\text{softplus}(s_\psi(x))+\varepsilon,
> > > $$
> > > where $s_\psi(x)$ is raw network output and $\varepsilon>0$ is small constant. To penalize excess precision, set regularizer $R_\psi(x)=(\gamma_\psi(x)-\beta)^2$ and define the training loss as
> > > $$
> > > \mathcal{L}(x,y)=\mathcal{J}_\psi(x,y)+\lambda R\_\psi(x),
> > > $$
> > > with $\lambda\geq0$ a tuning parameter.
> > > ## Experiments
> > > We follow official TorchUncertainty pipeline+UCI Kin8nm dataset (same settings on all methods). MAE/RMSE evaluate accuracy, NLL measures probabilistic fit, QCE evaluates uncertainty calibration.
> > >
> > > **Baseline** (NLL/QCE ) is point estimator-it does not model predictive uncertainty
> > >
> > > **Deep Evidential Regression (DER)** is unstable (NaN QCE) as per official TorchUncertainty DER tutorial
> > >
> > > **DAPPr** outperforms Ensemble on MAE RMSE and QCE. We use $\lambda=5e-4$ in the experiments
> > > Method|MAE↓|RMSE↓|NLL↓|QCE↓
> > > -|-|-|-|-
> > > Baseline|0.264|0.352|-|-
> > > +Normal|0.310|0.419|0.441|0.054
> > > +Laplace|0.320|0.438|0.486|0.067
> > > +Student|0.299|0.406|0.404|0.032
> > > +Cauchy|0.345|0.458|0.694|0.060
> > > MC Drop|0.496|0.619|0.935|0.033
> > > DER|0.301|0.410|**0.393**|NAN
> > > DAPPr|**0.261**|**0.347**|0.398|**0.012**
> > > -|-|Ensembles|-|-
> > > Ensemble|0.283|0.375|0.332|0.022
> > > Ensemble+DAPPr|**0.242**|**0.317**|**0.324**|**0.010**|
> > >
> > > # W3/W4/Q1: Explanation
> > > DAPPr achieves best or 2nd-best results on most metrics as:
> > > - **RSNN** performed better on CIFAR-100 (SVHN) due to **use of pretrained ResNet, borrowing ImageNet knowledge.**
> > > - **RKL-PN** performed better on StanfordDogs (DTD) as **it regularizes outputs for uniform noise, inducing bias toward texture datasets.**
> > > - **F-EDL** performs better on some OOD metrics due to **Spectral Normalization (SN).**
> > > - **R-EDL benefits from relaxed evidence offset.**
> > >
> > > **Incorporating these designs in DAPPr improves results:**
> > >
> > > StanfordDogs(DAPPr+Noise vs RKL-PN):
> > > Method|Acc|Conf|ImageNet-O|DTD|Places365
> > > -|-|-|-|-|-
> > > RKL-PN|50.2|80.1|95.2|82.9|69.6
> > > DAPPr+Noise|**61.7**|**88.1**|**96.1**|**84.2**|**80.1**
> > >
> > > CIFAR-100 and StanfordDogs (DAPPr+SN vs F-EDL):
> > > Method|Acc|Conf|SVHN|TinyImageNet
> > > -|-|-|-|-
> > > FEDL|69.4|94.0|75.3|80.5|
> > > DAPPr+SN|**70.9**|**94.9**|**76.5**|**81.7**
> > >
> > > Method|Acc|Conf|ImageNet-O|DTD|Places365
> > > -|-|-|-|-|-
> > > FEDL|53.0|82.9|95.1|82.5|70.6
> > > DAPPr+SN|**61.8**|**87.9**|**97.2**|**86.9**|**85.8**
> > >
> > > CUB-200-2011 (DAPPr+Relax vs R-EDL):
> > > Method|Acc|Conf|ImageNet-O|DTD|Places365
> > > -|-|-|-|-|-
> > > R-EDL|51.8|86.2|89.3|67.8|55.7
> > > DAPPr+Relax|**56.1**|**89.0**|**90.6**|**69.1**|**59.0**|
> > >
> > > - **When DAPPr is better:** In challenging settings with many similar competing classes. DAPPr suppresses spurious evidence of incorrect classes (Table 6). Its possibilistic posterior provides a principled uncertainty target that scales with data support, which other second-order predictors miss.
> > >
> > > - **When DAPPr is worse:** When others benefit from specific design choices.
> > >
> > > **DAPPr maintains simplicity/efficiency/outperforms others on average over 5 datasets:**
> > > Method|Acc|Conf|OOD
> > > -|-|-|-
> > > RKL-PN|60.7|86.6|70.1
> > > RSNN|64.0|88.7|75.6
> > > R-EDL|61.7|89.5|76.6|
> > > FEDL|61.2|85.0|76.0|
> > > DAPPr|**67.7**|**92.2**|**79.5**|
> > > Ensemble|69.3|91.8|78.9|
> > > Ensemble+DAPPr|**72.1**|**93.9**|**82.8**|
> > >
> > > DAPPr outperforms 2nd best second-order predictor by Acc/Conf/OOD of 3.6%/2.6%/2.9%, and improves ensembles by 2.8%/2.1%/3.8%.
> > >
> > > We will add this to revision.

---

### Official Review · Reviewer_n6qP · 2026-03-07

**Soundness:** 3
**Presentation:** 3
**Significance:** 3
**Originality:** 4
**Overall Recommendation:** 4
**Confidence:** 4

**Summary:**

This paper introduces DAPPr, an uncertainty quantification method grounded in possibility theory. Specifically, it defines a possibilistic posterior over parameters, projects it to the prediction space via a supremum operator, and approximates the resulting projected posterior using a Dirichlet possibility function, yielding a simple training objective. Overall, DAPPr offers a more theoretically grounded perspective on evidential uncertainty estimation and demonstrates strong empirical performance.

**Compliance With Llm Reviewing Policy:**

Affirmed.

**Final Justification:**

The authors provided a comprehensive rebuttal and follow-up comments that resolved most of my concerns and improved my understanding of the work. I find the paper to have a notable strength in originality, as, to my knowledge, possibility theory has not been widely explored in modern UQ research. However, some inherent limitations raised in my original review remain.

Overall, I am positive about the paper and believe it is sufficient for acceptance, but not strong enough to warrant an increased score. Therefore, I maintain my original score (4, Weak Accept).

**Key Questions For Authors:**

1. Why does DAPPr improve classification performance over the EDL baselines, and even over deep ensembles on fine-grained datasets? Given its design, I would have expected some trade-off in which improved epistemic uncertainty quantification comes at the cost of classification performance. Could the authors provide more intuition on where this performance gain comes from?

2. While the overall framework is motivated by possibility theory, the design of the regularizer, as well as the aleatoric and epistemic uncertainty measures, appears fairly standard. Could the authors discuss whether these components can also be derived more directly from possibility theory, rather than being adopted in a relatively conventional form?

3. How does DAPPr compare to non-EDL single-pass UQ methods? Since the paper is positioned as a broader epistemic UQ framework rather than merely an EDL variant, comparison to representative non-EDL baselines would strengthen the empirical positioning.

4. Can you provide ablation study results in the standard or long-tailed setting? This would help clarify how much each component contributes to the final performance, and whether the same design choices remain beneficial across different data regimes.

5. Is the proposed method related to [1], which likewise seems to study how a possibility-theoretic framework can capture both aleatoric and epistemic uncertainty?

In addition to these questions, it would be helpful if the authors could provide further clarification and justification regarding some of the concerns raised in the *Weaknesses* section.

**References**

[1] Hieu et al., Decoupling epistemic and aleatoric uncertainties with possibility theory, AISTATS 2025

**Limitations:**

Not explicitly. The paper would benefit from a more explicit discussion of its limitations, such as the fact that the underlying assumptions may not hold in low-capacity settings.

**Strengths And Weaknesses:**

### **Strengths**

* This paper offers a new perspective on single-pass uncertainty quantification (UQ) and EDL, which I find valuable, especially given the recent criticism that EDL lacks a solid theoretical foundation for epistemic UQ.
* The paper presents a theoretically grounded UQ framework based on the possibility theory.
* The experimental evaluation is comprehensive, and the empirical results are generally strong and convincing.


### **Weaknesses**

* While the possibilistic derivation is interesting, the final model obtained after the approximations appears quite close to a standard single-pass evidential predictor with complex parameterization. This makes it somewhat unclear whether the practical contribution lies in a fundamentally new UQ mechanism or primarily in a new theoretical interpretation of an otherwise fairly standard model.

* The main derivation relies on a strong approximation—namely, that imposing a prediction constraint on a single sample does not substantially affect the leave-one-out optimum. While this assumption may be plausible in highly overparameterized regimes, it is less clearly justified for finite-capacity models that may arise in practical settings.

* The methods and overall contributions are quite difficult to follow, partly because possibility theory is unfamiliar to many readers in the machine learning community. As a result, the paper can be hard to parse, and several key derivations and intuitions would benefit from more accessible explanations and clearer presentation.

---

> ### Author Rebuttal · Authors · 2026-03-29
>
> **We thank the Reviewer for helpful feedback.**
>
> # W1: Is it new UQ or just interpretation?
> While our model is lightweight/resembles a single-pass evidential predictor, **our contribution is not merely interpretive. The key distinction is whether epistemic uncertainty is formally defined & targeted.**
>
> - Epistemic uncertainty arises from lack of knowledge about the true parameters, which is fundamentally a parameter-space concept. Yet existing second-order predictors (e.g., EDL) bypass this origin. Epistemic uncertainty is never their target. **They assume a prediction-space distribution, fit labels, and infer uncertainty post-hoc from concentration.**
>
> - **DAPPr traces epistemic uncertainty to its source at parameter level via a possibilistic posterior, projects it to prediction space via supremum operators and then approximates the projected posterior with a Dirichlet for tractability. DAPPr thus learns to approximate a well-defined uncertainty target induced by parameter plausibility, a different mechanism from EDL.**
>
> Empirical results confirm it is not a reinterpretation: DAPPr's epistemic uncertainty decreases with more data (Fig. 2). Our regularizer is not interchangeable with EDL (Tab. 6).
> # W2: Assumption in finite-capacity regime.
> We agree the approximation is a regime assumption, not a universal identity.
>
> However:
> * **The overparameterized regime where our assumption holds is where epistemic uncertainty is most relevant:** when models fit training data well, the key issue is recognizing when predictions lack sufficient data support, especially on OOD inputs. In low-capacity settings, the priority is fit, not uncertainty.
> * Moreover, **modern deep nets are often overparameterized, and perturbing one sample has minimal effect on remaining data, implying the single-sample constraint has negligible impact on the leave-one-out optimum.**
>
>   **Appendix C supports this: the leave-one-out loss changes by only 0.25 relative to its scale 320 on CIFAR-100/ResNet-18.** We will clarify this/discuss tighter approx. for finite-capacity models as future work.
> # W3: Readers unfamilair with Possibility Theory (PT).
> Indeed, as PT is not mainstream, we will revise draft by adding more basics in our Supplementary, and improve accessibility by structuring our method in 4 steps:
> - Define the possibilistic posterior over parameters
> - Project it to prediction space via the supremum rule
> - Approximate the projected posterior with Dirichlet
> - Derive the closed-form surrogate under cross-entropy.
>
> **We will link each step to familiar probabilistic concepts and provide pseudocode (see our Resp. W3 to bGjM)**
> # Q1: Why DAPPr improves classification?
> - While one expects improved epistemic UQ hurts classification results, **our method does not simply reduce confidence uniformly.**
> - Instead, the possibilistic objective constrains the learned prediction-space possibility function relative to the projected posterior, while the regularizer suppresses spurious evidence on incorrect classes. **This acts as a structured regularizer on wrong-class evidence rather than a generic confidence penalty, improving discrimination, especially in fine-grained settings** where similar classes compete.
> - This is consistent with the empirical results in Tables 2, 3, 5.
> # Q2: Components derivation vs. adoption.
> Thank you. It is useful to distinguish first-principles components from design choices:
> - The possibilistic posterior, the projection to prediction space, and the $D_{\max}$-based matching objective are directly derived from possibility theory.
> - Dirichlet parameterization then yields a closed-form tractable implementation under cross-entropy.
> - By contrast, the spurious-evidence regularizer and scalar summaries used for aleatoric/epistemic uncertainty are best viewed as natural instantiations rather than unique axiomatic consequences.
>
> We will clarify this.
> # Q3: Non-EDL baselines.
> Kindly see our **Resp. W3&W4 to Ro7a for non-EDL comparisons on CIFAR/fine-grained/TinyImageNet,** and Resp. W4 to bGjM for UQ on LLM.
> # Q4: Ablations.
> Table (AUPR ↑) below shows that removing/replacing our regularizer with EDL's KL term degrades performance, confirming it is specific to the possibilistic formulation:
> ||CIFAR-10-LT| ($\rho$=0.01)||CIFAR-100|||
> :-|-:|:-|:-|-:|:-|:-
> Method|Acc.|Conf.|OOD|Acc.|Conf.|OOD
> DAPPr w/o Reg|66.74|83.35|57.49|64.45|90.52|71.52
> DAPPr w/ KL Reg|67.58|85.59|65.82|69.54|93.66|73.73
> EDL + our Reg|56.10|80.30|58.84|42.32|90.02|60.71
> DAPPr|**68.81**|**87.01**|**68.29**|**70.85**|**94.39**|**76.22**
> # Q5: Relation to [1] Hieu et al. (AISTATS'25).
> - Paper [1] is a general possibility-theoretic framework for jointly representing epistemic/aleatoric uncertainty/establishes theoretical foundations.
> - **DAPPr builds on [1] but pursues different goals:**
>   - deriving a tractable deep-learning method via possibilistic posterior projection
>   - Dirichlet approximation.
>
>   DAPPr thus introduces a set of new solutions. We will clarify this.

---

> > ### Author Rebuttal · Reviewer_n6qP · 2026-04-02
> >
> > I sincerely thank the authors for their comprehensive rebuttal, which helped me better understand the paper. While I like the paper and remain supportive of its acceptance, I do not think the paper’s significance is strong enough to justify increasing my score to 5.
> >
> > Therefore, I maintain my original score.

---

> > > ### Author Response · Authors · 2026-04-03
> > >
> > > We sincerely thank the reviewer for the kind and constructive engagement throughout the review process.
> > > \
> > > \
> > > **We apologize for not conveying the full strength of our work more clearly in the initial rebuttal.**
> > >
> > >
> > > ### 1. Below we summarize our key contributions and additional results that we hope strengthen the case for acceptance:
> > >
> > > - **Theoretical contribution: DAPPr is the first second-order predictor to formally define and target epistemic uncertainty at the parameter level via a possibilistic posterior, yielding a principled training objective with closed-form solutions.**
> > >   - Unlike existing second-order predictors, which construct their objectives directly in prediction space without a formal connection to parameter-level uncertainty, DAPPr derives its objective from a well-defined possibilistic posterior, providing the missing theoretical grounding.
> > >
> > > - **Empirical strength**: Below we average performance over 5 diverse datasets (CIFAR-10/100, CUB-200-2011, StanfordDogs, TinyImageNet) and compare with the most competitive second-order methods (RKL-PN/RSNN/R-EDL/FEDL):
> > >   Method|Acc|Conf|OOD
> > >   :-|:-:|:-:|:-:
> > >   RKL-PN|60.7|86.6|70.1
> > >   RSNN|64.0|88.7|75.6
> > >   R-EDL|61.7|89.5|76.6|
> > >   FEDL|61.2|85.0|76.0|
> > >   **DAPPr**|**67.7**|**92.2**|**79.5**|
> > >   Ensemble|69.3|91.8|78.9|
> > >   **Ensemble+DAPPr**|**72.1**|**93.9**|**82.8**|
> > >
> > >   In summary:
> > >   - **DAPPr outperforms all single-model second-order methods by 3.6%/2.6%/2.9% on Acc/Conf/OOD,** while matching CE in parameters, memory, training, and inference time.
> > >   - Notably, despite being a single model, **DAPPr surpasses a 10-model ensemble on confidence estimation and OOD detection,** confirming that principled uncertainty targeting yields stronger UQ than ensemble.
> > >  - **Combining DAPPr with ensembles further improves all metrics by 2.8%/2.1%/3.8%.**
> > >
> > > - **Generality of DAPPr:** DAPPr extends naturally to regression using a Gaussian possibility function under $L^2$ loss, admitting a closed-form surrogate.
> > >   \
> > >   \
> > >   On UCI Kin8nm, DAPPr outperforms all single-model baselines on MAE, RMSE, and QCE, including Deep Evidential Regression (DER) which produces invalid uncertainty estimates (NaN QCE):
> > > Method|MAE ↓|RMSE ↓|NLL ↓|QCE ↓
> > > :-|:-:|:-:|:-:|:-:
> > > Baseline|0.264|0.352|-|-
> > > +Normal|0.310|0.419|0.441|0.054
> > > +Laplace|0.320|0.438|0.486|0.067
> > > +Student|0.299|0.406|0.404|0.032
> > > +Cauchy|0.345|0.458|0.694|0.060
> > > MC Drop|0.496|0.619|0.935|0.033
> > > DER|0.301|0.410|**0.393**|NAN
> > > **DAPPr**|**0.267**|**0.347**|0.398|**0.012**
> > > -|-|Ensembles|-|-
> > > Ensemble|0.283|0.375|0.332|0.022
> > > **Ensemble+DAPPr**|**0.242**|**0.317**|**0.324**|**0.010**|
> > >
> > >
> > > - **Scalability**: DAPPr extends to LLMs where non-EDL methods are incompatible, outperforming other baselines including IB-EDL (ICLR 2025) on accuracy, NLL, and OOD detection on Mistral-7B fine-tuned on OBQA.
> > >
> > >
> > > ### 2. Compared to the well-accepted probabilistic framework, we acknowledge possibility theory is less familiar to the ML community.
> > >
> > > However:
> > > - **We believe this is precisely what makes this work significant**, just as Bayesian deep learning expanded the field by bringing probabilistic rigor to neural networks, we believe our work offers a complementary and equally rigorous foundation for epistemic uncertainty that the community has not yet explored.
> > > - **Our work uniquely enables what no existing framework achieves simultaneously:** principled epistemic uncertainty grounding AND computational tractability, resolving a fundamental dilemma that has persisted in the field.
> > > - By establishing this foundation, we open a new research direction with concrete building blocks for future work, expanding the theoretical boundary of uncertainty quantification beyond what probability theory alone can offer.
> > >
> > >
> > > We thank the Reviewer again for consideration. **Meantime, if you have any further questions we can answer/clarify, kindly do let us know.** While the system will not permit us post further answers (alas, ICML blocks them), **we will be sure to include them in the revised paper.**
> > > \
> > > \
> > > Kind regards,
> > > \
> > > Authors

---

### Official Review · Reviewer_SoMN · 2026-03-11

**Soundness:** 3
**Presentation:** 3
**Significance:** 3
**Originality:** 3
**Overall Recommendation:** 5
**Confidence:** 4

**Summary:**

This paper proposes the DAPPr framework for uncertainty estimation based on possibility theory. The design is motivated by quantifying epistemic uncertainty efficiently with second-order predictors in a more principled manner. The method begins with a possibilistic posterior in parameter space and derives a trainable Dirichlet-form predictor through projection and approximation in prediction space. Experiments on classification, OOD detection, and long-tailed data show competitive performance compared with several EDL baselines.

**Compliance With Llm Reviewing Policy:**

Affirmed.

**Final Justification:**

The rebuttal addressed my main concerns. This paper focuses on an important problem in reliable ML and offers a practical and novel perspective, with sufficient experimental support. Although the method still relies on some heuristic design, as also noted by other reviewers, I agree that these issues do not undermine the main contribution. I think it is worth accepting, and I’d like to increase my score from 4 to 5.

**Key Questions For Authors:**

(1) What is the essential difference between DAPPr and EDL?

(2) Aleatoric uncertainty reflects inherent randomness in the data. Why can it be estimated by $1−max_k​  α_k​/α_0​$ and used for confidence estimation in Tables 1–3?

(3) What is the essential cause of the empirical gains of DAPPr? Why should modeling uncertainty with possibility theory improve generalization? What optimization preferences does the proposed loss change?

**Limitations:**

The term “Predictive Uncertainty” in the title does not appear in the main text. Also, the relationship among predictive, epistemic, and aleatoric uncertainty is unclear. So it is hard to understand the paper’s main idea from the title alone.

The paper also lacks a clear discussion of limitations, for example, when the approximation in Eq.11 may fail, and which hard cases may be challenging.

**Strengths And Weaknesses:**

Soundness: The idea is interesting, and the theoretical story is also novel. However, the method is not fully principled, as it feels more like a theory-inspired method with several approximations. The experiments are sufficient, covering several tasks and settings.

Presentation: This paper is clearly written and well structured. But the link between the title and the main technical content could be made clearer.

Significance: This paper studies efficient and accurate technology for uncertainty quantification. This is an important problem for reliable machine learning and OOD detection. The method is tested across several tasks, suggesting decent practical value.

Originality: This paper gives a new perspective on using possibility theory for epistemic uncertainty quantification within an evidential learning framework. The originality is good, though not absolute.

---

> ### Author Rebuttal · Authors · 2026-03-30
>
> We thank the reviewer for the constructive comments.
>
> # Q1: Essential Difference between DAPPr and EDL.
> While DAPPr is lightweight and resembles a single-pass evidential predictor like EDL, they are very different. T**he key distinction** lies in whether epistemic uncertainty is ever formally defined and targeted in learning:
> - Epistemic uncertainty arises from lack of knowledge about the true parameters, which is fundamentally a parameter-space concept. Yet existing second-order predictors (e.g., EDL) bypass this origin. Epistemic uncertainty is never their target. **They assume a distributional family in prediction space, fit labels, and infer uncertainty post-hoc from concentration**.
> - DAPPr traces epistemic uncertainty to its source at the parameter level via a possibilistic posterior, projects it to prediction space via supremum operators and only then approximates the projected posterior with a Dirichlet for tractability.
>
>   **Unlike EDL, DAPPr explicitly approximates toward a well-defined uncertainty target** induced by parameter plausibility, a target EDL never establishes.
> - This theoretical difference has **direct empirical consequences: DAPPr's epistemic uncertainty consistently decreases with more data while EDL's pathologically increases (Figure 2)**
> - Moreover, **our regularizer is not interchangeable with EDL (Table 6)**, confirming the two frameworks are different.
>
> # Q2: Aleatoric Uncertainty for Confidence Estimation.
> - In classification, aleatoric uncertainty reflects the inherent ambiguity in $p(y|x)$: when multiple classes are equally likely under $p(y|x)$ at $x$, no model can predict the correct label reliably. High aleatoric uncertainty therefore directly bounds the probability of misclassification, making it a principled proxy for confidence estimation.
> - In DAPPr, The core possibilistic object is $g_\psi(\cdot|x)$ with mode $\alpha/\alpha_0$ and $\alpha_0$ controlling concentration around it. At the mode $\alpha_k/\alpha_0$ (the most plausible prediction), parameter uncertainty is set aside, any remaining ambiguity is irreducible by better parameters, and only comes from the data itself. Only aleatoric uncertainty remain. $1-\max_k\alpha_k/\alpha_0$ measures the remain uncertainty: small when one class dominates, large when classes compete, a natural possibilistic aleatoric uncertainty measure.
>
> Thus $1-\max_k\alpha_k/\alpha_0$ can be used for confidence estimation.
>
> # Q3: What causes classification gains?
> We do not claim possibility theory automatically improves generalization. **The gains stem from a specific training bias induced by the possibilistic derivation:**
> - **DAPPr does not simply reduce confidence uniformly. Two complementary mechanisms drive the gains:**
>   - **First**, ⁡$D_\max$ is one-sided, favoring minimal-commitment solutions that do not assign more plausibility than the projected posterior supports.
>   - **Second**, the regularizer suppresses spurious evidence on incorrect classes, **shifting the model preference from "fit the label with high evidence" to "fit the label while avoiding unsupported wrong-class sharpness."**
> - **Together, they act as a structured regularizer on wrong-class evidence rather than a generic confidence penalty, improving discrimination, especially in fine-grained settings where similar classes compete.**
> - The ablation confirms the gain is tied to the DAPPr objective-regularizer pair: removing/replacing our regularizer degrades performance (Table 6). Data-size analysis further shows DAPPr's epistemic uncertainty decreases with more data while EDL's pathologically increases (Figure 2). These results are consistent with Tables 2, 3 & 5.
>
> # L1: Predictive Uncertainty (paper title)
> We apologise: we will explicitly use "predictive uncertainty" in the introduction in revision.
> # L2: Relationship among predictive, epistemic, & aleatoric uncertainty.
> - **Predictive uncertainty** refers to total uncertainty in the model's prediction.
> - **Aleatoric uncertainty** captures the inherent noise in the data (e.g., class overlap or ambiguous inputs) and is irreducible even with more data.
> - **Epistemic uncertainty** captures uncertainty about the model parameters due to limited data and can be reduced as more data is observed.
>
> **Predictive uncertainty can be decomposed into aleatoric & epistemic components. DAPPr models them both through the Dirichlet possibility.**
> # L3: When Eq. (11) fails?
> We will add a dedicated limitations paragraph in revision:
> - Approximation in Eq. (11) relies on the high-capacity assumption and is most appropriate in the overparameterized regime where modern deep networks naturally reside, e.g., ResNets, VGGs.
> - In low-capacity models (such as linear classifiers), improving fit is the priority and uncertainty quantification is secondary, making the approximation less relevant.
> - An explicit failure mode is already noted in Section 4.3: conflicting labels at the same input.

---

> > ### Author Rebuttal · Reviewer_SoMN · 2026-04-02
> >
> > Thank you for the clarification.
> >
> > I still maintain that it is conceptually inappropriate to use aleatoric uncertainty to estimate the model’s confidence.
> >
> > As the paper is good in other aspects, and after considering other reviewers’ comments, I would maintain my original rating for this paper.

---

> > > ### Author Response · Authors · 2026-04-03
> > >
> > > We sincerely thank the reviewer for the prompt follow-up and for the insightful comments.
> > >
> > >
> > >
> > > - **We do agree with Rev's concern that the Aleatoric uncertainty and model confidence are conceptually different**, and relating them requires careful justification.
> > > - We would like to clarify that our choice simply follows a widely adopted operational practice in probabilistic classification and uncertainty quantification, which is not an assumption specific to our work or introduced by us.
> > >
> > > In standard classification, models output a predictive distribution $p(y \mid x)$, which is routinely interpreted as the model’s confidence in its prediction (e.g., via maximum class probability). Importantly, this predictive distribution inherently reflects data (aleatoric) uncertainty, as it captures ambiguity in the input–label relationship.
> > >
> > > **Thus, many works use quantities derived from the predictive distribution as practical measures of confidence:**
> > >
> > > - Kendall & Gal (2017, What Uncertainties Do We Need in Bayesian Deep Learning for Computer Vision?) distinguish aleatoric and epistemic uncertainty, and model aleatoric uncertainty directly in the predictive likelihood; the resulting predictive distribution is used to assess prediction confidence in practice.
> > > - Hüllermeier & Waegeman (2019, Aleatoric and Epistemic Uncertainty in Machine Learning: An Introduction to Concepts and Methods) describe aleatoric uncertainty as uncertainty inherent in the conditional distribution ( $p(y \mid x)$ ), which is precisely the quantity used for probabilistic predictions.
> > > - Gawlikowski et al. (2022, A Survey of Uncertainty in Deep Neural Networks) note that class probabilities provide a direct probabilistic interpretation and form the basis for commonly used uncertainty (and confidence) measures.
> > >
> > > **These works do not claim a strict equivalence between aleatoric uncertainty and confidence.** Rather, they support the view that predictive probabilities (which encode aleatoric uncertainty) are routinely used as a practical proxy for confidence in classification settings:
> > >
> > > - Our work follows this convention for comparability with prior methods and evaluation protocols, not because we assume a strict conceptual equivalence. We agree that this distinction should be made explicit, and we will revise the paper to clearly separate aleatoric uncertainty, epistemic uncertainty, and confidence.
> > >
> > > **We sincerely appreciate your comment, as it highlights an important conceptual issue relevant to the broader UQ community.**
> > >
> > > We will be sure to clarify this point in the revised manuscript. **Meantime, if there is anything more we can clarify/explain further, do not hesitate to let us know and we will revise accordingly the paper**.
> > >
> > > Kind regards,
> > > \
> > > Authors

---

### Official Review · Reviewer_bGjM · 2026-03-24

**Soundness:** 2
**Presentation:** 3
**Significance:** 3
**Originality:** 3
**Overall Recommendation:** 4
**Confidence:** 4

**Summary:**

Authors propose a novel method for uncertainty estimation in classification based on possibility theory approach to statistical inference. Relying on possibilistic analogue of amortized Bayesian inference, approximate the resulting predictive possibility function with a neural network. To avoid bi-level optimization they propose a tractable loss surrogate. Extensive experiments compare their approach with existing baselines and Evidential Deep learning-based methods.

**Compliance With Llm Reviewing Policy:**

Affirmed.

**Final Justification:**

The paper presents a new perspective on Dirichlet-based uncertainty estimation methods, connecting them with possibility theory and inference. Although not ground-breaking, this work's theoretical and practical contributions will be of interest to the uncertainty estimation community ICLR.

**Key Questions For Authors:**

1. Can you provide a possibilistic interpretation of the uncertainty measures and the regularizer that you use?
 2. Can you comment on the partial order argument and why inner optimization problem of eq 8 will not cause the degeneracy?
 3. How exactly is the loss in eq 17 implemented? Do you stop the gradients at the $\tilde{p}*$?

**Limitations:**

yes

**Strengths And Weaknesses:**

I was not very familiar with possibility theory, but the general premise and motivation, up to about Section 4.2 seem sound to me. However, the the remaining part that discusses the derivation of the surrogate objective is a bit problematic. First, it is not clear while having only partial order will not lead to under-estimation of the actual possibility values (being too conservative) in inner problem $D_{max}$, eq 8. Authors state that an uninformative initialization of the parameters prevents this, but what exactly this initialization is is not discussed. The supplied argument is also not convincing since the effects of different input affect all trained parameters simultaneously. Second, The final practical method still relies on several heuristic choices not derived from possibility theory (Section 4.4): using $\mathbf{\alpha} = \mathrm{softplus}(\Phi'_{\psi}(x))+1$ and regularizer in eq 16.
A clear prseudocode of the actual implemented loss will enhance the presentation of the results and clarify potential questions. The evaluation is extensive but too focused on just EDL variants.

---

> ### Author Rebuttal · Authors · 2026-03-30
>
> # W1: Initialization and shared parameters.
> - **Re. initialization:** "uninformative initialization" in the paper refers to $\alpha_0=0$, i.e., full uninformativeness where $g_\psi(p|x)=1$ for all $p$.
>   - While our implementation requires $\alpha_k>1$ for tractability (Prop. 1) and cannot achieve $\alpha_0=0$, **standard random network initialization produces nearly equal $\alpha_k$ across classes, so $g_\psi$ achieves its maximum 1 at the uniform prediction.**
>   - Since $g_x^\*$ is peaked near the true label and does not assign 1 there, $g_\psi$ over-estimates $g_x^\*$ at training start.
>   - The inner maximization finds $p\*$ near the uniform prediction, the outer minimization reduces $g_\psi$ at $p\*$, progressively pushing the mode of $g_\psi$ toward that of $g_x^\*$.
>
>   **Thus, no special initialization is required in practice.**
> - **Re. shared parameters:**
>   - Shared parameters couple inputs which is precisely the reason Sec. 4.3 introduces the high-capacity approximation: in the overparameterized regime of modern deep nets, the network can satisfy any single-sample constraint without substantially affecting predictions on other inputs, thus decoupling the interaction.
>   - This is a regime assumption, not a universal identity: Appendix C empirically validates it.
>
> We will clarify this.
> # W2: Heuristic choices.
> We agree the final implementation includes design choices beyond the core possibilistic derivation, but neither is arbitrary:
> - Constraint $\alpha=\text{softplus}(\Phi_\psi(x))+1$ is a direct consequence of Prop. 1: the closed-form surrogate maximizer $\tilde{p}^*$ lies in the simplex only when $\alpha_k>1$, **making this a necessary implementation requirement, not a mere arbitrary choice.**
> - Regularizer in Eq. (16) prevents unbounded spurious evidence under the per-sample surrogate objective, consistent with the minimum commitment principle of possibility theory. We will clarify it in the revision.
> # W3&Q3: Pseudocode.
> We stop gradient for $\tilde{p}^*$.
> ```
> Input: x, y, net Φ'_ψ, λ
>  α ← softplus(Φ'_ψ(x))+1
>  α_0 ← ∑_k α_k
>  p̃* ← stopgrad((α - y)/(α_0 - 1))
>  log_g ← α_0·log(α_0) + ∑_k α_k·log(p̃*_k / α_k)
>  R ← ||(1 - y) ⊙ α||²
>  loss ← log_g + λ·R
> ```
> # W4: Beyond EDL baselines.
> Besides non-EDL comparisons on CIFAR/fine-grained/TinyImageNet (see **Resp. W3&W4 to Ro7a**), DAPPr's simplicity enables extension to LLMs where non-EDL methods are computationally costly or incompatible.
> \
> \
> **As in IB-EDL (ICLR 2025), we finetune Mistral-7B on OBQA and report OOD AUROC↑ on (ARC-C, ARC-E, CSQA). DAPPr outperforms them:**
> Method|Acc.↑|ECE↓|NLL↓|ARC-C|ARC-E|CSQA
> :-|:-:|:-:|:-:|:-:|:-:|:-:
> CE|88.06|11.29|0.85|60.40|53.30|63.70|
> MC Drop|88.07|11.22|0.84|60.39|53.30|63.70|
> Ensemble|88.51|8.87|0.67|60.67|54.05|63.80|
> EDL|87.23|6.23|0.47|77.34|74.18|78.28|
> I-EDL|88.06|9.63|0.45|82.28|79.42|82.07|
> R-EDL|88.33|5.43|0.41|72.85|67.56|71.93|
> IB-EDL|88.73|**2.27**|0.41|88.58|94.29|83.85|
> DAPPr|**89.47**|3.11|**0.36**|**90.42**|**95.12**|**90.19**
> # Q1: Possibilistic Interpretation.
> The core possibilistic object is $g_\psi(\cdot|x)$ with mode $\alpha/\alpha_0$ and $\alpha_0$ controlling concentration around it.
> - **Aleatoric uncertainty $1-\max_k\alpha_k/\alpha_0$:** at the mode $\alpha_k/\alpha_0$ (the most plausible prediction), parameter uncertainty is set aside, any remaining ambiguity is irreducible by better parameters, and only comes from the data itself. $1-\max_k\alpha_k/\alpha_0$ measures this: small when one class dominates, large when classes compete, a natural possibilistic aleatoric uncertainty measure.
> - **Epistemic uncertainty $K/\alpha_0$:** when $\alpha_0$ is small, $g_\psi$ is broad, assigning high plausibility to many predictions and approaching total ignorance $\boldsymbol{1}$ as $\alpha_0$→0. $K/\alpha_0$ measures this proximity to total ignorance, directly reflecting insufficient parameter knowledge, the defining characteristic of epistemic uncertainty.
> - **Regularizer:** possibility theory's minimum commitment principle prescribes that plausibility should not exceed what the evidence supports. The regularizer enforces this by suppressing spurious evidence on incorrect classes, plausibility contradicted by the observed label.
> # Q2: Partial order/degeneracy in Eq. (8)
> We believe there is a misunderstanding due to differences between possibility functions and probability densities:
> - Unlike densities, **possibility functions are normalized to a maximum of 1, not an integral of 1, meaning $\sup_pg_\psi(p|x)=1$ must hold at all times.**
> - **Collapse or degeneracy thus cannot occur by construction:** Since $g_\psi$ always assigns 1 at its mode $\alpha/\alpha_0$, and $g_x^\*$ does not assign 1 there unless their modes coincide, $g_\psi$ always over-estimates $g_x^\*$ at its mode, the inner optimization always finds a meaningful point of over-estimation, driving $g_\psi$'s mode toward that of $g_x^\*$. Once their modes coincide, partial order is satisfied, the loss vanishes.
>
> We will clarify it.

---

> > ### Author Rebuttal · Reviewer_bGjM · 2026-04-04
> >
> > I thank the authors for the detailed response to my feedback and providing experimental results. The concerns were addressed and I increase my confidence and maintain the positive score.

---

> > > ### Author Response · Authors · 2026-04-05
> > >
> > > Esteemed Reviewer,
> > > \
> > > \
> > > We sincerely thank the Reviewer for the careful and constructive engagement throughout the review process, and for taking the time to reconsider our work after the rebuttal.
> > > \
> > > \
> > > We are glad that our clarifications and additional results have addressed the concerns and improved the Reviewer's confidence in the approach.
> > > \
> > > \
> > > **Meantime, if you have any further questions or suggestions, kindly let us know.**
> > > \
> > > \
> > > **Rest assured, we will incorporate all discussed improvements, results and explanations. We will deliver a strong and polished final version of our paper.**
> > > \
> > > \
> > > Kind regards,
> > > \
> > > Authors

---

### Decision · Program_Chairs · 2026-04-30

**Decision:**

Accept (regular)

**Comment:**

This paper proposes the DAPPr framework for uncertainty estimation based on possibility theory. The design is motivated by quantifying epistemic uncertainty efficiently with second-order predictors in a more principled manner. Experiments on classification, OOD detection, and long-tailed data show competitive performance compared with several baselines.

This paper focuses on predictive uncertainty modeling for deep learning, which is a crucial problem. This paper offers a new perspective (at least to many machine learning researchers) from the possibility theory, which I find valuable. The experimental evaluation is comprehensive, and the empirical results are strong and convincing.

All reviewers recommend accepting this paper. After reading the paper, the reviews, and the rebuttals, I agree with the reviewers and recommend accepting this paper.